# Single-cell transcriptomic analysis identifies the conversion of zebrafish Etv2-deficient vascular progenitors into skeletal muscle

Brendan Chestnut[1,4], Satish Casie Chetty[1,4], Andrew L. Koenig 🄳 [1,2,4] & Saulius Sumanas 🄳 [1,3✉]

Cell fate decisions involved in vascular and hematopoietic embryonic development are still poorly understood. An ETS transcription factor Etv2 functions as an evolutionarily conserved master regulator of vasculogenesis. Here we report a single-cell transcriptomic analysis of hematovascular development in wild-type and *etv2* mutant zebrafish embryos. Distinct transcriptional signatures of different types of hematopoietic and vascular progenitors are identified using an *etv2^ci32Gt* gene trap line, in which the Gal4 transcriptional activator is integrated into the *etv2* gene locus. We observe a cell population with a skeletal muscle signature in *etv2*-deficient embryos. We demonstrate that multiple *etv2^ci32Gt*; *UAS:GFP* cells differentiate as skeletal muscle cells instead of contributing to vasculature in *etv2*-deficient embryos. Wnt and FGF signaling promote the differentiation of these putative multipotent *etv2* progenitor cells into skeletal muscle cells. We conclude that *etv2* actively represses muscle differentiation in vascular progenitors, thus restricting these cells to a vascular endothelial fate.

[1] Division of Developmental Biology, Cincinnati Children's Hospital Medical Center, 3333 Burnet Ave, Cincinnati, OH 45229, USA. [2] Center for Cardiovascular Research, Washington University School of Medicine, 660S. Euclid Ave, St. Louis, MO 63110, USA. [3] Department of Pediatrics, University of Cincinnati College of Medicine, Cincinnati, OH 45229, USA. [4] These authors contributed equally: Brendan Chestnut, Satish Casie Chetty, Andrew L. Koenig.
✉email: saulius.sumanas@cchmc.org

D uring embryonic development, the lateral plate mesoderm (LPM) gives rise to multiple different cell lineages which include vascular endothelial cells, hematopoietic lineages and cardiomyocytes[1,2]. The specification of these lineages occurs in parallel to the specification of adjacent skeletal muscle progenitors which are thought to originate in the somites from different progenitors than the LPM lineages[3]. Although many factors involved in different steps of LPM lineage specification and differentiation have been identified, the entire timeline, transitional steps, and changes in the global transcriptional program that occur during hematopoietic and cardiovascular differentiation in vivo are still poorly understood.

While it is challenging to study early cardiovascular development in mammalian embryos, the zebrafish has emerged as an advantageous model system to study early cell fate decisions during embryogenesis. The signaling pathways and transcriptional programs that regulate the specification and differentiation of the LPM lineages are highly conserved between zebrafish and other vertebrates[4].

We and others have previously identified an ETS transcription factor Etv2/Etsrp which is one of the earliest markers of vascular and hematopoietic progenitor cells and functions as a key regulator of vascular and hematopoietic development in multiple vertebrates including mouse and zebrafish[5–7]. In zebrafish embryos, etv2 is expressed in vascular endothelial progenitor cells, as well as early myeloid and erythroid progenitors, and its expression is downregulated after cells undergo hematopoietic and vascular differentiation[5,6]. In the absence of Etv2 function, vascular endothelial and myeloid progenitors fail to differentiate. While some of them undergo apoptosis, others can acquire alternative cell fates and differentiate into cardiomyocytes, demonstrating fate flexibility of early progenitors[8–10].

The relatively recent emergence of highly efficient and high-throughput single-cell transcriptomic technologies has facilitated extensive probing of cellular diversity and complex cell differentiation pathways both in vitro and in vivo. In recent years, several studies have been performed to delineate the transcriptional diversity of vascular cell types, and to uncover lineage commitment trajectories during cardiovascular development[11–13]. However, the fate decisions of LPM-derived cells are still poorly understood.

Here, we report single-cell transcriptomic profiling of etv2-expressing cells in wild-type and etv2-deficient zebrafish embryos during early stages of vascular development. We identify the transcriptional profiles and novel transitional states of cells during different steps of vascular and hematopoietic differentiation. Our results show that in the absence of Etv2 function, vascular progenitors can acquire a skeletal muscle fate, arguing that Etv2 function is required to actively repress alternative cell fates in multipotent mesodermal progenitors. These findings will be important in understanding the ontogeny of different cardiovascular and hematopoietic lineages and will help in designing more efficient in vitro and in vivo cell differentiation strategies to generate different types of progenitors for therapeutic purposes.

## Results

**Single-cell RNA-seq of vascular and blood progenitors**. To analyze the diversity of hematovascular progenitors, we performed single-cell RNA-seq of GFP-positive cells sorted by fluorescence-activated cell sorting (FACS) from heterozygous and homozygous etv2^ci32Gt; UAS:GFP zebrafish embryos at the 20-somite stage. This reporter line, recently generated by CRISPR mediated homology-independent repair, has an insertion of the gal4 reporter within the etv2 coding sequence[14]. As described previously, heterozygous etv2^ci32Gt embryos recapitulate the endogenous expression pattern of etv2 in vascular endothelial

progenitors and differentiated vascular endothelial cells, while homozygous embryos show profound defects in vascular development due to the interruption of the etv2 coding sequence[14] (Supplementary Fig. 1).

Transcriptomes of 2049 and 588 cells were obtained from heterozygous and homozygous etv2^ci32Gt embryos, respectively, using the Chromium system (10× Genomics) which employs a microdroplet technology to isolate individual cells, followed by the next-generation sequencing. The relative frequency of GFP+ cells out of the total number of cells was similar in heterozygous and homozygous embryos (1.89% and 1.98%, respectively). Transcriptomes from heterozygous and homozygous embryos were pooled and clustered using Seurat[15], resulting in 12 distinct cell clusters which were visualized using the t-distributed stochastic neighbor embedding (t-SNE) approach[16] (Fig. 1a–d). We subsequently assigned cell identities based on marker genes which were significantly enriched in each cluster (Supplementary Table 1, Supplementary Datas 1 and 2). Two different clusters (#2 and #3) corresponded to vascular endothelial cells and were thus labeled as EC1 and EC2. The EC1 cluster showed expression of multiple known vascular endothelial markers, including cdh5, cldn5b, egfl7, sox7, ecscr and others, while the top genes expressed in EC2 cells included crip2, tpm4a, fli1a, and erg, all known to label vascular endothelial cells (Supplementary Table 1 shows the top marker genes for each cluster, while t-SNE and violin plots for selected marker genes are shown in Fig. 1c and Supplementary Figs. 2 and 3). There was a large overlap in marker expression between the EC1 and EC2 groups. It is currently unclear whether EC1 and EC2 cells represent distinct cellular identities or different stages of endothelial differentiation. Cluster #4 corresponded to the vascular endothelial progenitor group (EPC) which showed enriched expression of tal1, lmo2, etv2, tmem88a, egfl7 (Fig. 1c, d, Supplementary Figs. 2 and 4, Supplementary Table 1). Although some of these genes (tal1, lmo2, etv2) are known to label both vascular and hematopoietic progenitors[5,17,18], other markers specific to this group (egfl7, sox7, fli1b) label vascular and not hematopoietic cells[19–21], arguing that this population corresponds to vascular endothelial progenitors. Two groups of cells with a strictly hematopoietic gene signature were identified. Cluster #7 showed specific expression of cebpb, spi1b, cebpa, cxcr3.2, all known to be specifically enriched in macrophages (Fig. 1c, d, Supplementary Figs. 2 and 5, Supplementary Table 1, Supplementary Data 1). Cluster #11 showed strong expression of multiple hemoglobin genes, including hbbe3, hbbe1.3, hbae1.3, as well as klf17, blf and other genes that are specific to red blood cells (Fig. 1d, Supplementary Figs. 2 and 5, Supplementary Table 1, Supplementary Data 1). Although etv2 in zebrafish does not show significant expression in zebrafish blood cells, etv2:GFP expression has been previously observed in myeloid and erythroid cells[22], likely due to the expression of etv2 in hematopoietic progenitors, which becomes downregulated as they differentiate. Cluster #10 had very few significantly enriched genes, which included a novel protein si:ch211–11n16.3, and genes tubb2b and hmg2b which are likely to be ubiquitously expressed. Apoptosis and cell cycle regulators pmaip1/noxa and ccng1[23,24] also showed relatively strong expression in this cell population (Fig. 1d, Supplementary Fig. 6, Supplementary Table 1, Supplementary Data 1), suggesting that this group is likely to include pro-apoptotic cells, which were observed in etv2^ci32Gt embryos. Previous studies have established that the loss of etv2 function results in increased apoptosis of vascular or hematopoietic cells[21,25]. Cluster #12 showed strong expression of hoxa11b, hoxb7a, cdx4, apoeb, which are all expressed in the posterior mesenchyme either in zebrafish or mouse embryos[26–28] (Fig. 1c, d, Supplementary Figs. 2 and 6, Supplementary Table 1, Supplementary Data 1). The tailbud progenitors have been shown to include multipotent cells which can give rise to endothelial, somitic and neural lineages[29]. Therefore this group may

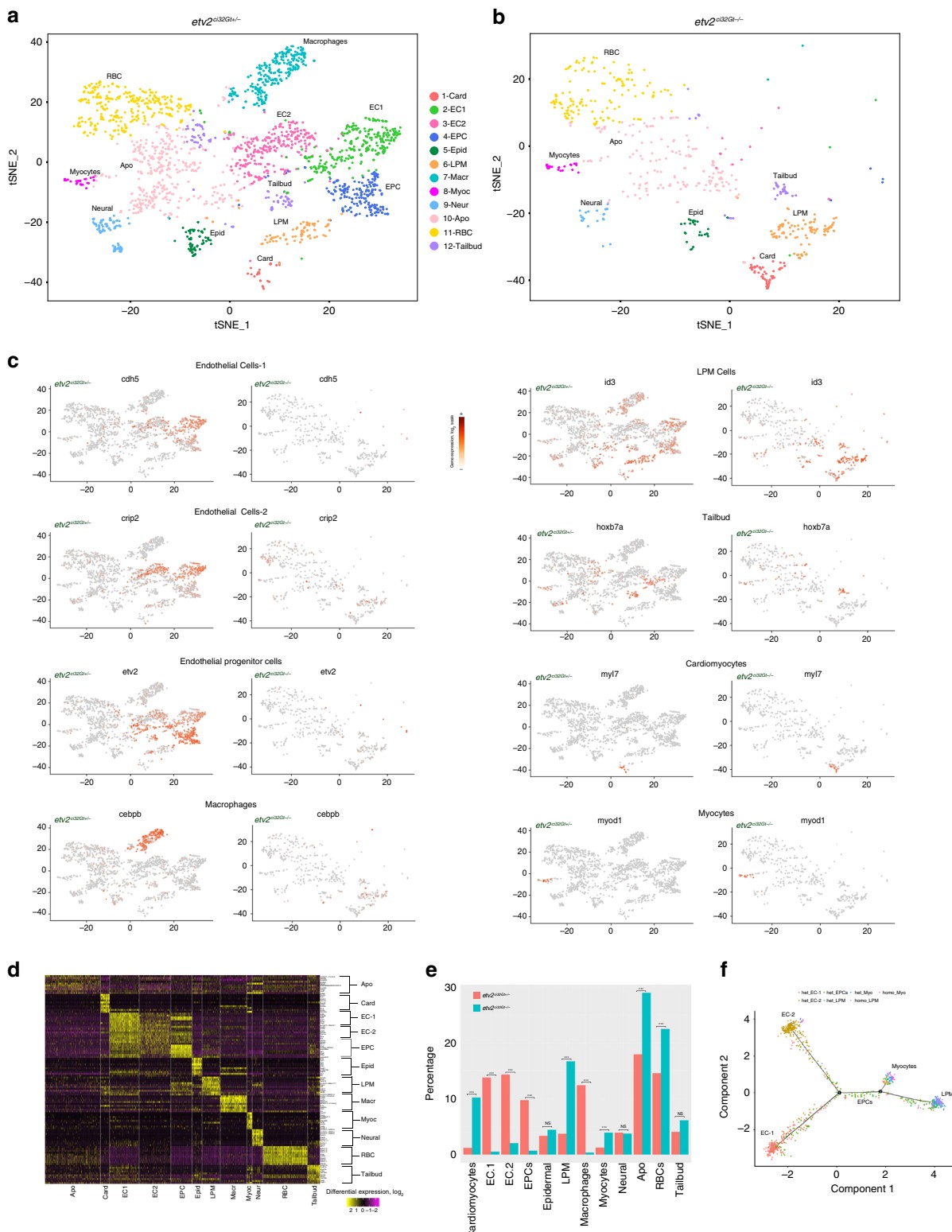

include early stage endothelial progenitors located in the tailbud. Indeed, many cells from the tailbud population exhibit noticeable expression of EPC markers (Fig. 1d, Supplementary Fig. 4). Another intriguing population of *etv2+* cells (cluster #6) showed expression of BMP response genes *id3* and *id1*, as well as *twist1a*, *pitx3*, and *prrx1a* (Fig. 1c, d, Supplementary Figs. 2 and 4, Supplementary Table 1, Supplementary Data 1). Our subsequent analysis (see below) demonstrated that all analyzed genes exhibited expression in

the LPM area; therefore we annotated this cluster as LPM cells. Additional cell populations identified in our analysis included cells with neural, epidermal, cardiomyocyte, and skeletal muscle-specific gene signatures (Fig. 1c, d, Supplementary Figs. 2, 7, and 8, Supplementary Table 1, Supplementary Data 1). Observation of GFP-positive cardiomyocyte and skeletal muscle populations in both *etv2^{ci32Gt}* heterozygous and homozygous embryos was somewhat unexpected. Cardiomyocytes in wild-type embryos do not

**Fig. 1 Single-cell RNA-seq analysis of *etv2*[ci32Gt]; *UAS:GFP* heterozygous and homozygous embryos at the 20-somite stage. a, b** t-SNE plots and cell clustering analysis. Twelve different clusters were identified. Card cardiomyocytes, EC endothelial cells, EPC endothelial progenitor cells, Epid epidermal cells, LPM lateral plate mesoderm, Macr macrophages, Myoc myocytes, Neur neural, Apo putative pro-apoptotic cells, RBC red blood cells, Tailbud tailbud progenitors. **c** t-SNE plots showing selected top markers for different cell types. **d** Heatmap of marker gene expression in different cell populations. **e** The proportions of GFP + cell types in *etv2*[ci32Gt+/−] and *etv2*[ci32Gt−/−] embryos. Note a great reduction in EC1, EC2, EPC, and macrophage populations and an increase in LPM, cardiomyocyte, myocyte, RBC, and Apo populations in *etv2* mutants. ***$p < 0.001$, NS not significant, chi-square test. $p$ Values: Card—$5.0 \times 10^{-26}$, EC1— $6.2 \times 10^{-18}$, EC2—$1.8 \times 10^{-14}$, EPCs—$2.4 \times 10^{-11}$, Epid—0.66, LPM—$8.4 \times 10^{-27}$, Macr—$2.5 \times 10^{-16}$, Myoc—$3.4 \times 10^{-4}$, Neur—1.0, Apo—$1.9 \times 10^{-7}$, RBC—$1.1 \times 10^{-4}$, Tailbud—0.21. Totally, 2049 and 588 cells total from *etv2*[ci32Gt+/−] and *etv2*[ci32Gt−/−] embryos, respectively, were analyzed in a single scRNA-seq experiment. **f** Pseudotime analysis graph of cells in LPM, EPC, EC1, EC2, and myocyte populations in *etv2*[ci32Gt] homozygous and heterozygous embryos.

express *etv2*, while our previous study has shown that *etv2*-deficient endothelial progenitors can differentiate as cardiomyocytes[8]. The presence of GFP-positive cardiomyocytes in *etv2*[ci32Gt] heterozygous embryos could be explained by the loss of one functional *etv2* allele, which resulted in some *etv2*+ cells differentiating as cardiomyocytes. However, differentiation of hematovascular progenitors into skeletal muscle cells has not been previously reported.

**scRNA seq analysis of *etv2* loss-of-function embryos**. We then analyzed the cell clusters present in *etv2*[ci32Gt] homozygous embryos, which exhibit severe defects in vascular differentiation. Both EC1 and EC2 populations, EPCs and macrophages were nearly completely absent in *etv2*[ci32Gt] homozygous embryos (Fig. 1b, e), consistent with the known requirement of *etv2* in the formation of these lineages[5,6]. In contrast, percentage of RBC cells was slightly but significantly increased in *etv2*[ci32Gt] homozygous embryos (Fig. 1b, e). To confirm this result, we analyzed expression of RBC-specific markers using data from the recently reported global transcriptome study of *etv2*[ci32Gt] heterozygous and homozygous embryos[14]. Indeed, expression of multiple RBC-specific genes, including *hbaa1, vwf, hbae3* and others was significantly upregulated in *etv2*[ci32Gt] homozygous embryos (Supplementary Table 2). Furthermore, differential expression analysis of scRNA-seq data between RBC cell populations in heterozygous and homozygous embryos revealed an upregulation of several RBC-specific genes in homozygous embryos, including *alas2, hbae3, hbae1.3,* and *hbbe1.1*, while several vascular endothelial specific genes including *tpm4, ecscr* and an EPC specific gene *lmo2* were downregulated (Supplementary Data 3). For additional confirmation, we knocked down *etv2* function using previously validated MO[5] in *gata1:dsRed* transgenic embryos which express fluorescent reporter in RBCs. *Etv2* knockdown embryos were morphologically normal and showed previously reported vascular defects that were similar to the *etv2* mutant phenotype. Subsequently, cells were disaggregated at 23 hpf and analyzed by FACS sorting. A statistically significant increase in the number of dsRed-positive cells was observed in *etv2* knockdown embryos (Supplementary Fig. 9). This suggests that the erythropoiesis pathway is upregulated in *etv2*-deficient embryos, although exact mechanism requires further investigation.

The tailbud, epidermal and neural populations were not significantly affected in *etv2*-deficient embryos. Interestingly, the LPM, cardiomyocyte, and skeletal muscle (myocyte) populations were greatly increased in the homozygous embryos (Fig. 1b, e). While an increase in cardiomyocyte population was expected because *etv2*+ cells are known to differentiate as cardiomyocytes in *etv2*-inhibited embryos[8], the presence of a myocyte population was intriguing. These results suggested that some *etv2*-deficient vascular progenitors may acquire skeletal muscle fate. Pseudotime analysis of EPCs, ECs, LPM cells and myocytes showed the distribution of EPCs which appear to originate from the cluster of LPM cells and progress along a differentiation trajectory towards either EC-1 or EC-2 states, while the myocyte pool branches off from the main trajectory at an earlier time point (Fig. 1f).

Cells in the homozygous embryos showed an increase in the LPM and myocyte populations, and a loss of EC populations. We hypothesized that the LPM group represented EPCs at an earlier stage of differentiation, and that the myocyte cells were an alternative cell fate for *etv2*+ progenitors when they could not differentiate into the vascular endothelial lineage.

**Identification of an endocardial subcluster**. In an attempt to identify additional heterogeneity within the endothelial cells, which may be missed in the global cell clustering approach, we performed subclustering of endothelial cell populations EC1 and EC2 identified in *etv2*[ci32Gt+/−] embryos using Seurat (see Methods). The subclustering of EC1 resulted in the identification of two subpopulations of cells (Fig. 2a–c), while the subclustering of EC2 did not yield meaningful subpopulations. One of the two EC1 sub-clusters showed enriched expression of *cdh6, gata5, fn1a,* and *drl*, among the other markers (Fig. 2d, e, Supplementary Data 4). *gata5* has been previously known to be enriched in endocardial cells[30]. In situ hybridization (ISH) analysis confirmed that expression of *fn1a* and *drl* was also enriched in the endocardial cells and largely absent from other vascular endothelial cells (Fig. 2f, g). In contrast, genes enriched in the remaining EC1 subpopulation (designated as EC1a) which included *etv2* and *lmo2*, were largely absent from endocardial progenitors at the 20-somite stage (Fig. 2h, i). This strongly suggests that the first subcluster represents a population of endocardial progenitors which possess a unique transcriptomic signature even prior to the formation of the heart tube.

**Vascular progenitors convert into muscle in *etv2* mutants**. To confirm if GFP expression was in fact present in skeletal muscle cells, as suggested by scRNA-seq analysis, we analyzed *etv2*[ci32Gt]; *UAS:GFP* embryos by confocal microscopy. Indeed, GFP expression was observed in several myocytes in *etv2*[ci32Gt] heterozygous embryos ($3 \pm 2.5$ myocytes per embryo, ± refers to standard deviation, $n = 8$ embryos), while multiple GFP-positive cells ($18.5 \pm 5.4$ per embryo, $n = 10$) were present within the somitic muscle of *etv2*[ci32Gt] homozygous embryos at 25 hpf (Fig. 3a–e). These cells displayed an elongated shape, similar to other skeletal muscle cells, and were positive for expression of skeletal muscle actin *actc1b:GFP* (Fig. 3f–h). GFP expression in skeletal muscle overlapped with antibody staining for fast muscle but not slow muscle cells (Supplementary Fig. 10). When crossed into the previously generated *etv2:mCherry* reporter line[31], these muscle cells were also positive for mCherry expression (Fig. 3i–k), indicating that the reporter fluorescence is unlikely to be an artifact of transgene misexpression in unrelated tissues. To confirm that *etv2*[ci32Gt]; *UAS:GFP* expression in the skeletal muscle is caused by the loss of *etv2* function, we crossed the *etv2*[ci32Gt] line with *etv2*[ci33] loss-of-function mutants. The *etv2*[ci33] allele, generated using CRISPR/Cas9 mutagenesis, contains a 13 bp insertion in exon 5 of the *etv2* gene (see "Methods"). Homozygous *etv2*[ci33] mutants show vascular defects similar to the previously reported *etv2*[y11] allele, which is likely to be null[21] (Supplementary Fig. 11a–d).

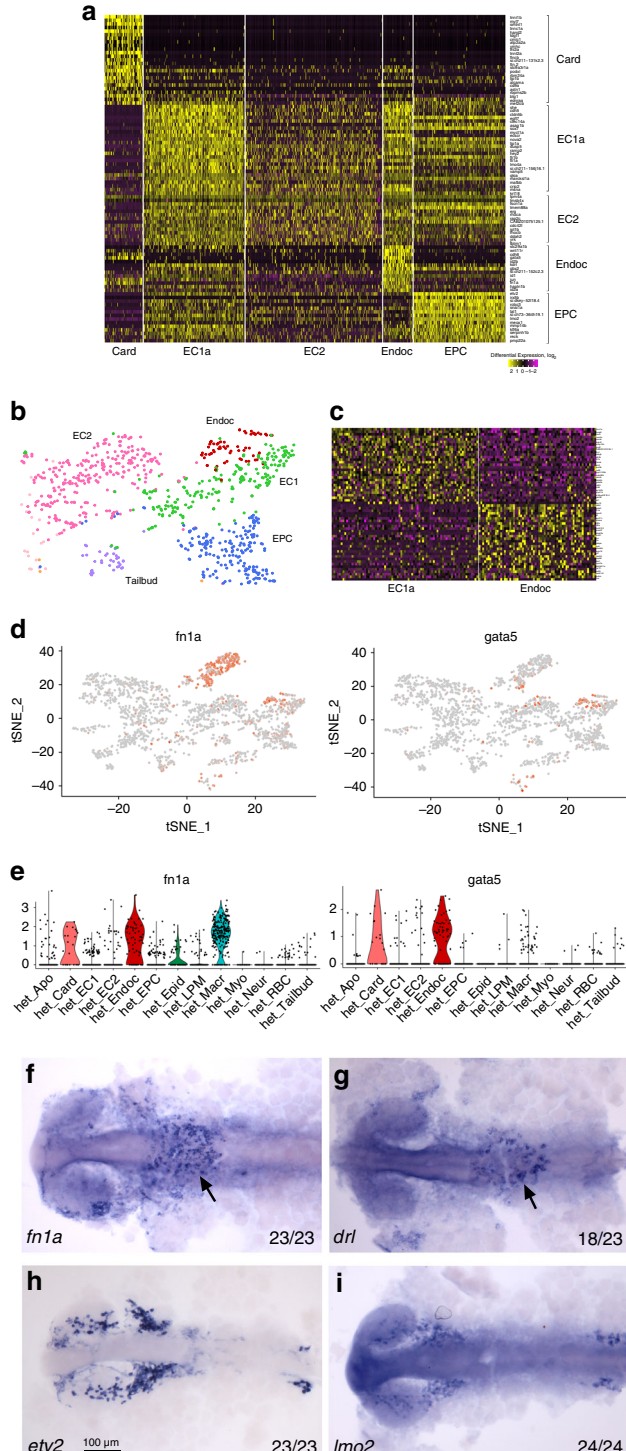

**Fig. 2 Identification of the transcriptional signature of endocardial progenitors.** Subclustering of the EC1 population in *etv2*<sup>ci32Gt+/−</sup> embryos identified a novel endocardial subpopulation (while the remaining endothelial cells are designated as EC1a). **a** A section of a heatmap showing marker gene expression in endocardial, endothelial (EC1a and EC2), endothelial progenitor (EPC), and cardiomyocyte populations. **b** A portion of the global t-SNE plot showing the endocardial subpopulation. **c** Heatmap for subclustering of EC1 population showing genes enriched in endocardial and the remaining endothelial (EC1a) cells. **d**, **e** t-SNE and violin plots for selected endocardial genes. **f–i** In situ hybridization analysis at the 20-somite stage for selected endocardial-enriched genes *fn1a* and *drl* (**f**, **g** arrows point to endocardial expression) and endocardial-excluded genes *etv2* and *lmo2* (bilateral expression is present in cranial endothelial progenitors but is largely absent from the endocardial cluster). Flat-mounted embryos, ventral view, anterior is to the left. The number of embryos displaying the representative phenotype out of the total number of embryos obtained from two replicate experiments is shown.

To investigate the mechanism of the conversion of these *etv2*-deficient progenitors into skeletal muscle in greater detail, we analyzed the functional role of bHLH transcription factor *scl/tal1* which functions downstream of *etv2* during vascular development and myelopoiesis[6]. Previous research has demonstrated that the combined knockdown of *scl* and *etv2* results in ectopic myocardial differentiation of *etv2+* progenitors in zebrafish[10], and that *scl* represses myocardial differentiation in mouse embryos[33]. However, its role in repressing skeletal muscle differentiation has not been investigated. The number of myocytes, positive for *etv2*<sup>ci32Gt+/−</sup>; *UAS:GFP* expression, was significantly increased in the *etv2*<sup>ci32Gt+/−</sup> embryos injected with the previously validated *scl* morpholino (MO)[33], compared to the control *etv2*<sup>ci32Gt+/−</sup> embryos (Fig. 3n, p). Furthermore, these ectopic myocytes were positive for expression of the skeletal muscle actin *actc1b* (Fig. 3q-s), arguing that *scl* knockdown results in increased skeletal muscle differentiation of *etv2*<sup>ci32Gt+/−</sup>; *UAS:GFP* cells. However, injection of *scl* mRNA into *etv2*<sup>ci32Gt</sup>; *UAS:GFP* embryos did not have a significant effect on the number of ectopic GFP-positive muscle cells (Supplementary Fig. 12), suggesting that *etv2* and *scl* genes may not function in a simple linear pathway to repress muscle differentiation in *etv2+* progenitor cells.

Heat-shock inducible *etv2* expression has been demonstrated to convert differentiated skeletal muscle into vascular endothelial cells[34]. To test if *etv2* is sufficient to inhibit early specification or differentiation of myocytes, we overexpressed *etv2* mRNA in zebrafish embryos. We have previously shown that such over-expression induces strong ectopic expression of vascular endothelial markers[5]. Conversely, expression of the early regulators of muscle differentiation *myod*, *myf5*, and *myog* was greatly inhibited upon *etv2* overexpression (Fig. 3t, u).

Vascular endothelial progenitors are known to originate in the LPM region. However, the somitic origin of vascular progenitors has also been suggested[35]. In order to analyze the origin of GFP+ myocytes, we performed time-lapse imaging of *etv2*<sup>ci32Gt</sup>; *UAS:GFP* embryos. The majority of EPCs in heterozygous embryos originated in the LPM region and subsequently migrated to the midline where they coalesced into the DA and the PCV (Fig. 4a–f, Supplementary Fig. 13a–f, Supplementary Movies 1 and 2). Occasional cells failed to migrate to the midline and elongated to form somitic muscle. In contrast, GFP+ cells in homozygous embryos failed to migrate to the midline. Some of these cells underwent apoptosis while others elongated and incorporated into the somitic muscle at approximately 14–18-somite stages (Fig. 4g–m, Supplementary Fig. 13g–l,

Double heterozygous *etv2*<sup>ci32Gt/ci33</sup> embryos displayed a significant increase in GFP+ myocytes compared to *etv2*<sup>ci32Gt+/−</sup> embryos (Fig. 3l–n, Supplementary Fig. 11e–h). In addition, GFP expression in skeletal muscle cells was also observed in *etv2*<sup>yl1−/−</sup> mutant embryos crossed into the previously established *Tg(-2.3 etv2:GFP)* reporter line but not in wild-type *Tg(-2.3 etv2:GFP)*[14,32] embryos (Fig. 3o). Collectively, these results argue that *etv2*-expressing progenitors differentiate as skeletal muscle in the absence of Etv2 function.

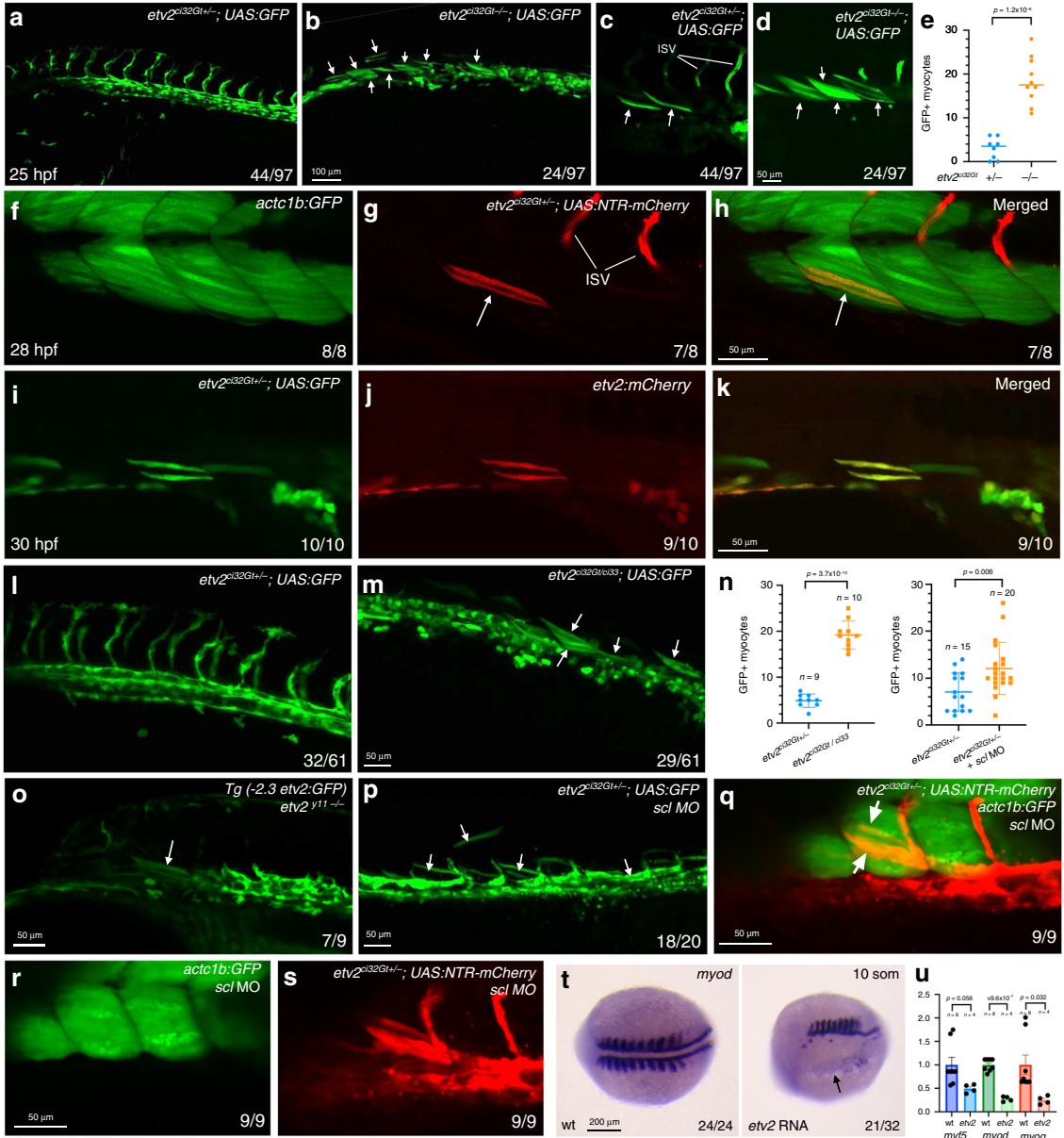

**Fig. 3 *Etv2*-expressing cells differentiate as skeletal muscle cells in the absence of *etv2* function. a–d** Trunk region of *etv2^ci32Gt^; UAS:GFP* embryos at 25 hpf. Maximum intensity projections of selected confocal slices are shown in (**c, d**). Note the absence of intersegmental vessels (ISVs) and elongated GFP-positive skeletal muscle fibers (arrows, **c, d**). The embryos were obtained from an incross of heterozygous *etv2^ci32Gt^* carriers; embryo numbers (lower right) correspond to the expected Mendelian ratio. **e** Quantification of GFP+ myocytes in the trunk region of 8 *etv2^ci32Gt+/−^* and 10 *etv2^ci32Gt−/−^*; UAS:GFP embryos at 25 hpf, analyzed in 2 replicate experiments. The bars show median values. **f–h** Co-expression of *etv2^ci32Gt+/−^; UAS:NTR-mCherry* and muscle-specific *actc1b:GFP* in mCherry-positive muscle cells (arrows, **g, h**). **i–k** The *Tg(etv2:mCherry)* line shows mCherry expression in skeletal muscle cells when crossed to *etv2^ci32Gt+/−^; UAS:GFP* carriers. **l, m** Multiple GFP+ skeletal muscle cells are apparent in the progeny of *etv2^ci32Gt+/−^; UAS:GFP* zebrafish crossed with the *etv2^ci33+/−^* line, which carries a loss-of-function mutation in *etv2*. Note that the expected frequency of double heterozygous embryos in (**m**) is 50%. **n** Quantification of GFP+ myocytes in the trunk region of *etv2^ci32Gt/ci33^* and *etv2^ci32Gt+/−^; scl MO* embryos shown in (**m, p**) obtained in two replicate experiments. The graphs show median and SD values. **o** GFP+ skeletal muscle cells observed in *etv2^y11−/−^* embryos crossed into the *Tg(-2.3 etv2: GFP)* reporter line. **p** Multiple GFP + skeletal muscle cells are apparent in *etv2^ci32Gt+/−^; UAS:GFP* embryos injected with *scl* MO. See graph **n** for quantification. **q–s** Ectopic myocytes observed in *scl* MO-injected *etv2^ci32Gt+/−^; UAS:NTR-mCherry* embryos are positive for muscle-specific *actc1b:GFP* expression at 24 hpf. **t** *etv2* RNA overexpression inhibits *myod* expression (arrow). Dorsal view, anterior is to the left. **u** qPCR analysis of *myf5*, *myod*, and *myog* expression in *etv2* RNA-injected and uninjected control embryos at the 10-somite stage. Mean values ± SEM is shown. RNA was purified from groups of ten embryos analyzed in two replicate experiments. In all graphs, two-tailed Student's *t* test was used. The number of embryos displaying the representative phenotype out of the total number of embryos obtained from two replicate experiments is shown.

Supplementary Movies 3–5). Thus, *etv2^ci32Gt^; UAS:GFP* cells that give rise to the skeletal muscle cells in the homozygous embryos appear to be derived from the initial pool of *etv2+* cells in the LPM region. To analyze when GFP+ positive cells initiate muscle marker

expression, we performed fluorescent ISH for *myod* expression combined with GFP fluorescence analysis at the 8–10-somite stages in *etv2^ci32Gt+/−^; UAS:GFP* embryos. GFP and *myod* co-expressing cells were positioned at the posterior edge of the somites next to

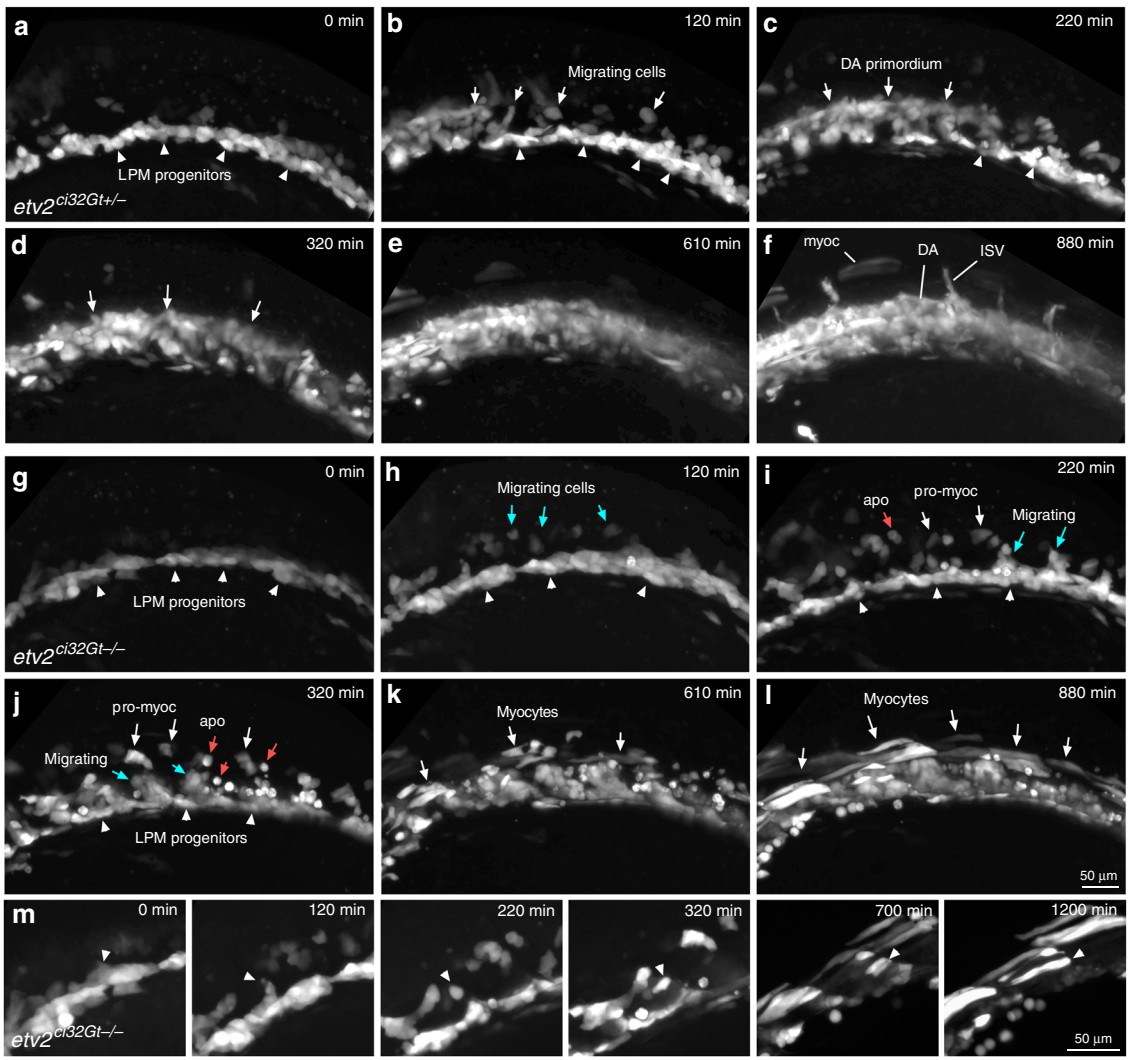

**Fig. 4 Time-lapse imaging of cell migration in *etv2^ci32Gt* embryos starting at the 9–10-somite stage.** Lateral view is shown, anterior is to the left. **a–f** In *etv2^ci32Gt+/−*; *UAS:GFP* embryos, bilaterally located vascular and hematopoietic progenitors within the lateral plate mesoderm (LPM, arrowheads) migrate toward the midline and coalesce into the axial vasculature (arrows). Note that some cells remain in the lateral position and elongate into muscle cells (myoc). DA progenitors of the dorsal aorta, ISV intersegmental vessels. Time frames are selected from the Supplementary Movie 1. **g–l** In *etv2^ci32Gt−/−*; *UAS:GFP* embryos, cells initiate migration (blue arrows) but fail to coalesce into the axial vasculature. Instead, many cells either undergo apoptosis (apo, red arrowheads point to round apoptotic cells) or differentiate into myocytes (white arrows, myoc). Time frames are selected from the Supplementary Movie 3. **m** Higher magnification view showing differentiation of a GFP+ progenitor cell initially positioned in the LPM into a myocyte (arrowhead points to the same cell). Note that the cell migrates dorsally from the LPM into the somite and then elongates as it undergoes differentiate into the muscle. Time frames are selected from the Supplementary Movie 4. Representative embryos are shown out of the total of seven heterozygous and four homozygous *etv2^ci32Gt*; *UAS:GFP* embryos that were imaged in two replicate experiments.

GFP+ cells in the LPM (Fig. 5). In wild-type embryos, *etv2*-expressing vascular progenitors are known to migrate between the somites and coalesce into axial vasculature starting at the 8–10-somite stage[36]. In contrast, some GFP+ cells migrate into the somites instead, and initiate *myod* expression in *etv2^ci32Gt* embryos.

**Wnt and FGF pathways are required for ectopic muscle.** Wnt and FGF signaling pathways have been previously implicated in somitic differentiation and myogenesis[37]. We tested if these pathways were involved in the differentiation of *etv2^ci32Gt*; *UAS: GFP*+ progenitors into ectopic muscle cells. To inhibit Wnt signaling we induced expression of Wnt inhibitor *dkk1* at the 8-somite stage. As it has been previously demonstrated[29], Wnt inhibition at this stage resulted in reduced *myod* expression in the tailbud region, while *myod* expression in differentiated somitic cells

was largely unaffected (Supplementary Fig. 14a, b). Heat-shock inducible expression of Wnt inhibitor *dkk1* greatly reduced the number of *etv2*+ myocytes observed in heterozygous *etv2^ci32Gt*; *UAS:GFP* embryos (Fig. 6a–c). Dkk1 overexpression did not affect the total number of cells positive for the UAS reporter expression in *etv2^ci32Gt* embryos (Supplementary Fig. 14c). This suggests that Wnt signaling promotes muscle differentiation in multipotent *etv2*+ progenitors.

To investigate the role of FGF signaling, we treated *etv2^ci32Gt−/−* embryos with SU5402, a chemical inhibitor of FGF signaling[38]. SU5402-treated embryos showed a significant decrease in the number of ectopic GFP+ muscle cells (Fig. 6d–f). To confirm the requirement of FGF signaling in the ectopic muscle differentiation, *etv2^ci32Gt+/−*; *UAS:NTR-mCherry* carriers were crossed with a heat-shock inducible *hsp70:dnFGFR1* line[39], and the expression of

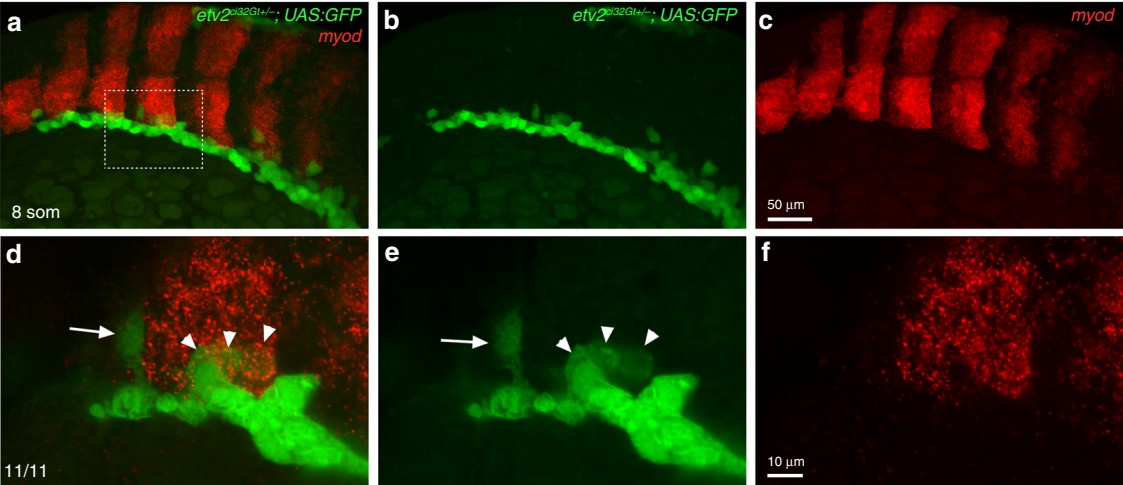

**Fig. 5 Fluorescent in situ hybridization for *myod* expression in *etv2*^ci32Gt+/−*; UAS:GFP* embryos.** In situ hybridization was performed using hybridization chain reaction (HCR) at the 8–10-somite stages. **a–c** Maximum intensity projection is shown; **d–f** An area boxed in **a** was imaged at higher magnification; maximum intensity projection of three confocal slices is shown. Note that most GFP-expressing cells are positioned in the lateral plate mesoderm, while some cells are starting to migrate toward the midline (arrow, **d**, **e**, points to a migrating angioblast which does not have *myod* expression). GFP and *myod* co-expressing cells (arrowheads, **d**, **e**) are apparent at the posterior edge of the somite. The punctate *myod* expression pattern is due to the nature of the HCR probe. Dorsolateral view, anterior is to the left. The numbers in the lower left corner display the number of embryos showing the phenotype out of the total number of embryos analyzed in two replicate experiments.

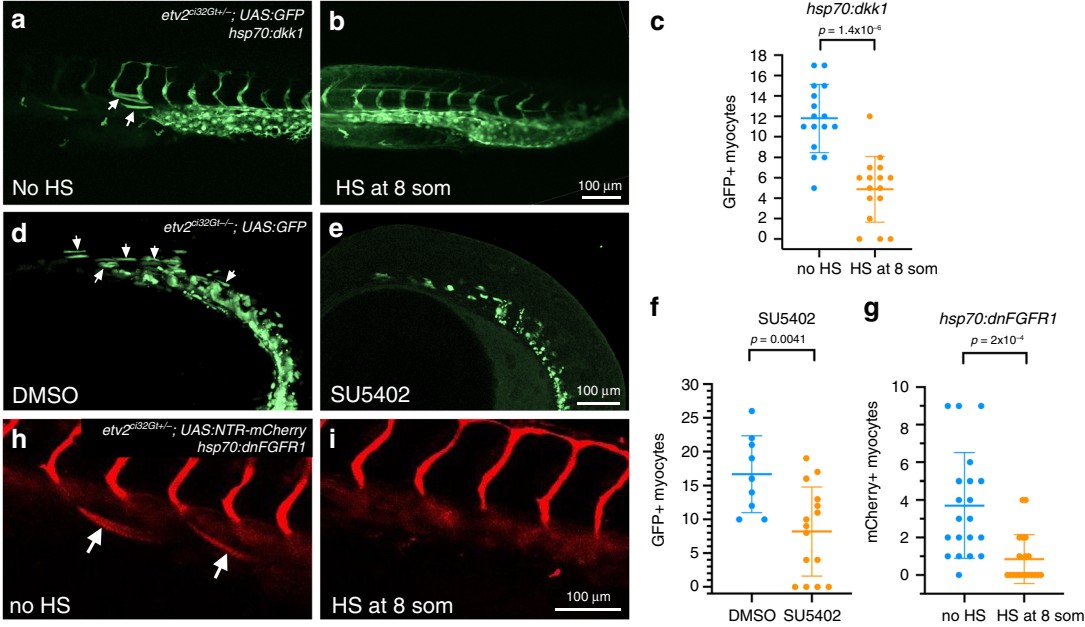

**Fig. 6 Wnt and FGF signaling is required for differentiation of *etv2*^ci32Gt*; UAS:GFP* cells into skeletal muscle. a–c** Heat-shock inducible expression of Wnt inhibitor Dkk1 greatly reduces GFP-positive muscle cells in *hsp70:dkk1; etv2*^ci32Gt+/−*; UAS:GFP* embryos. Heat-shock (HS) was performed at the 8-somite stage, and GFP fluorescence imaged at 24–26 hpf. Sixteen embryos were analyzed in each group in two replicate experiments. **d–f** *etv2*^ci32Gt−/−*; UAS:GFP* embryos treated with FGF inhibitor SU5402 starting at the 50% epiboly stage show a greatly reduced number of GFP-positive cells at the 22-somite stage. Nine control DMSO-treated and 15 SU5402-treated embryos were analyzed in two replicate experiments. **g–i** Heat-shock inducible expression of dnFGFR1 results in a significant reduction of mCherry-positive muscle cells in *etv2*^ci32Gt+/−*; UAS:NTR-mCherry; hsp70:dnFGFR1* embryos. Twenty each control and HS *hsp70:dnFGFR1* embryos were analyzed in two replicate experiments. Median ± SD values are shown in all graphs, two-tailed Student's *t* test was used for statistical analysis.

dnFGFR1 was induced at the 8-somite stage. A significant reduction in the number of ectopic muscle cells was observed in FGF-inhibited embryos while no significant defects in vascular development were apparent (Fig. 6g–i). The overall number of UAS reporter-positive cells was not affected in FGF-inhibited *etv2*^ci32Gt+/− embryos (Supplementary Fig. 14c). Together, these results suggest that FGF signaling is required for muscle-specific differentiation of multipotent *etv2*+ progenitors.

**Identification of putative multipotent LPM progenitors.** Based on scRNA-seq analysis, cell cluster #6 was significantly increased in *etv2^{ci32Gt−/−}; UAS:GFP* embryos. Marker genes for this population included BMP response genes *id3* and *id1* (Supplementary Table 1, Supplementary Data 1), which have been recently shown to function upstream of *etv2* during vascular differentiation of tailbud-derived neuromesodermal progenitors[40]. Another marker gene for this population, *twist1a*, has been implicated in developmental angiogenesis[41], although its mechanistic role in this process has not been investigated. Other genes enriched in this cluster included transcription factors *prrx1a*, *foxd1*, and *foxd2*, which have been implicated in diverse processes, including epithelial–mesenchymal transition and craniofacial development[42,43]. However, their function or relevance to vascular development has yet to be elucidated. We decided to investigate the identity of the cells from cluster #6 in greater detail. Expression of transcription factors *twist1a*, *prrx1a*, and *foxd2* was present in the LPM region at the 18-somite stage (Fig. 7a–n). Based on ISH analysis of *etv2^{ci32Gt+/−}; UAS:GFP* embryos, *prrx1a* and *twist1a* expression was absent from the midline GFP+ cell population which correspond to vascular endothelial cells that are coalescing into axial vessels, while it partially overlapped with GFP expression in the most lateral vascular progenitors. The expression of these genes was not significantly affected in *etv2^{ci32Gt−/−}; UAS: GFP* embryos (Fig. 7d–f, j–n). Similar overlap of *prrx1a* or *twist1a* expression with GFP fluorescence in the most lateral vascular progenitors was also observed in wild-type *TgBAC(etv2:GFP)* embryos (Supplementary Fig. 15). Because vascular progenitors originate within the LPM prior to their migration to the midline, such gene expression pattern suggests that this cell cluster corresponds to the LPM cells which may include the earliest vascular progenitors (Fig. 7o, p).

**Transcriptional profiling of *etv2:GFP* cells using Fluidigm.** Although the Chromium-based scRNA-seq analysis identified novel cell populations, we were not able to distinguish endothelial subpopulations such as arterial and venous progenitors which are known to separate early during vasculogenesis. We decided to supplement our analysis by sequencing fewer cells at a higher coverage. For this approach, we used a previously established *Tg(-2.3 etv2:GFP)* line[31] which shows specific GFP expression in vascular endothelial cells (Fig. 8a–c). Cells from embryos at 16–20-somite stages were dissociated and sorted using the Fluidigm C1 system. Totally, 96-cells were subjected to scRNA-seq analysis. After cell clustering using AltAnalyze software[44], 8 distinct cell populations were identified. They included macrophage, erythroid, tailbud progenitor, and nonspecific epidermal cell populations (Fig. 8d–g). Four different vascular endothelial cell populations were identified. The vascular endothelial progenitor population displayed high expression of *etv2*, *lmo2* and *tal1*, as well as multiple genes that have not been previously associated with EPCs (Fig. 8d–g, Supplementary Datas 5 and 6). Many genes overlapped between EPC populations identified in the Chromium and Fluidigm analyses, suggesting that the same populations of cells were identified using both approaches. Cluster 5 was enriched in venous-specific vascular endothelial genes such as *flt4* and *stab1l*, while cluster 6 showed high expression of known arterial-specific markers, including *dll4*, *efnb2a*, *cldn5b* (Fig. 8g, Supplementary Data 5). Cluster 7 included many genes with known expression in vascular endothelium such as *col4a1*, *ldb2a*, or *hapln3*. Validation by ISH analysis confirmed expected expression patterns of EPC, arterial and venous markers and showed that expression of the cluster 7 marker *ldb2a* was also primarily enriched in the arterial vasculature (Fig. 8i–l). The biological difference between the

clusters 6 and 7 is unclear, and it will require further investigation. Notably, marker genes for cluster 7 (also labeled as EC2) are largely different from the EC2 cluster identified using the Chromium platform.

**Expression of venous markers in arterial progenitors.** Previous studies have suggested that arterial and venous progenitors originate at distinct spatial regions and at different times[36]. However, it has been controversial as to when cells acquire distinct arterial and venous identities[45–47]. We used the expression of well-established arterial and venous markers to calculate an arteriovenous index for each vascular endothelial cell (see "Methods"). While some cells showed a very clear arterial or venous signature, many cells co-expressed both arterial and venous markers (Fig. 8h). Fluorescent ISH analysis for venous *dab2* and arterial marker *cldn5b* expression confirmed that many arterial progenitors co-express both markers at the 20-somite stage (Fig. 8m–o). In contrast, arterial *dab2* expression becomes downregulated by 24 hpf and is restricted to the PCV (Fig. 8p–r). Intriguingly, ventrally located progenitors of the PCV mostly displayed venous marker expression and showed very little, if any, arterial *cldn5b* expression (Fig. 8m–o). Thus, arterial progenitors co-express both arterial and venous markers at early stages of vascular development while venous progenitors show largely venous-specific marker expression.

**Discussion**

In this study we used scRNA-seq analysis to identify transcriptional signatures and investigate transitional states of cell populations during the differentiation of vascular endothelial and hematopoietic lineages. Our results suggest that cells in the LPM are multipotent and that the same progenitors can give rise to hematopoietic, vascular, myocardial, and even skeletal muscle lineages. Wnt and FGF signaling pathways promote skeletal muscle development in *etv2*-negative multipotent progenitors (Fig. 9). Etv2 overexpression was sufficient to inhibit muscle differentiation and instead promoted hematovascular fates. Our results argue that in addition to activating vasculogenic and hematopoietic programs, *etv2* also actively represses myogenesis, possibly through the transcription factor *scl/tal1*. Previous work has demonstrated that *etv2* can similarly repress myocardial differentiation in both zebrafish and murine embryos[8,9]. Scl, similar to Etv2, also represses myocardial fates by occupying primed enhancers of myocardial specific genes and preventing their activation by cardiac factors[48]. It is tempting to speculate that Etv2 also inhibits muscle specification by a similar mechanism, where Scl functions downstream of Etv2 and binds to muscle-specific promoters, thus preventing their activation by Myod. Alternatively, Scl and Etv2 may form a protein complex and/or function in parallel pathways to repress myocardial differentiation. Because *scl* overexpression failed to repress the formation of ectopic myocytes observed in *etv2*-deficient embryos, we speculate that *scl* may not function in a simple linear pathway downstream of *etv2* to inhibit myogenesis. While the precise molecular pathways remain to be established, they will likely involve epigenetic mechanisms to repress the transcriptional activation of the myogenic program when the hematovascular program is activated.

A recent study has demonstrated two opposing activities of FGF and BMP signaling in specifying somitic progenitors and the LPM[40]. FGF signaling through bHLH factors *myf5*, *myod* and *msgn1* induced medial fate such as skeletal muscle in tailbud-derived neuromesodermal progenitors while BMP signaling induced blood and endothelium marker expression, and *etv2* expression in particular, through transcriptional activation of *id*

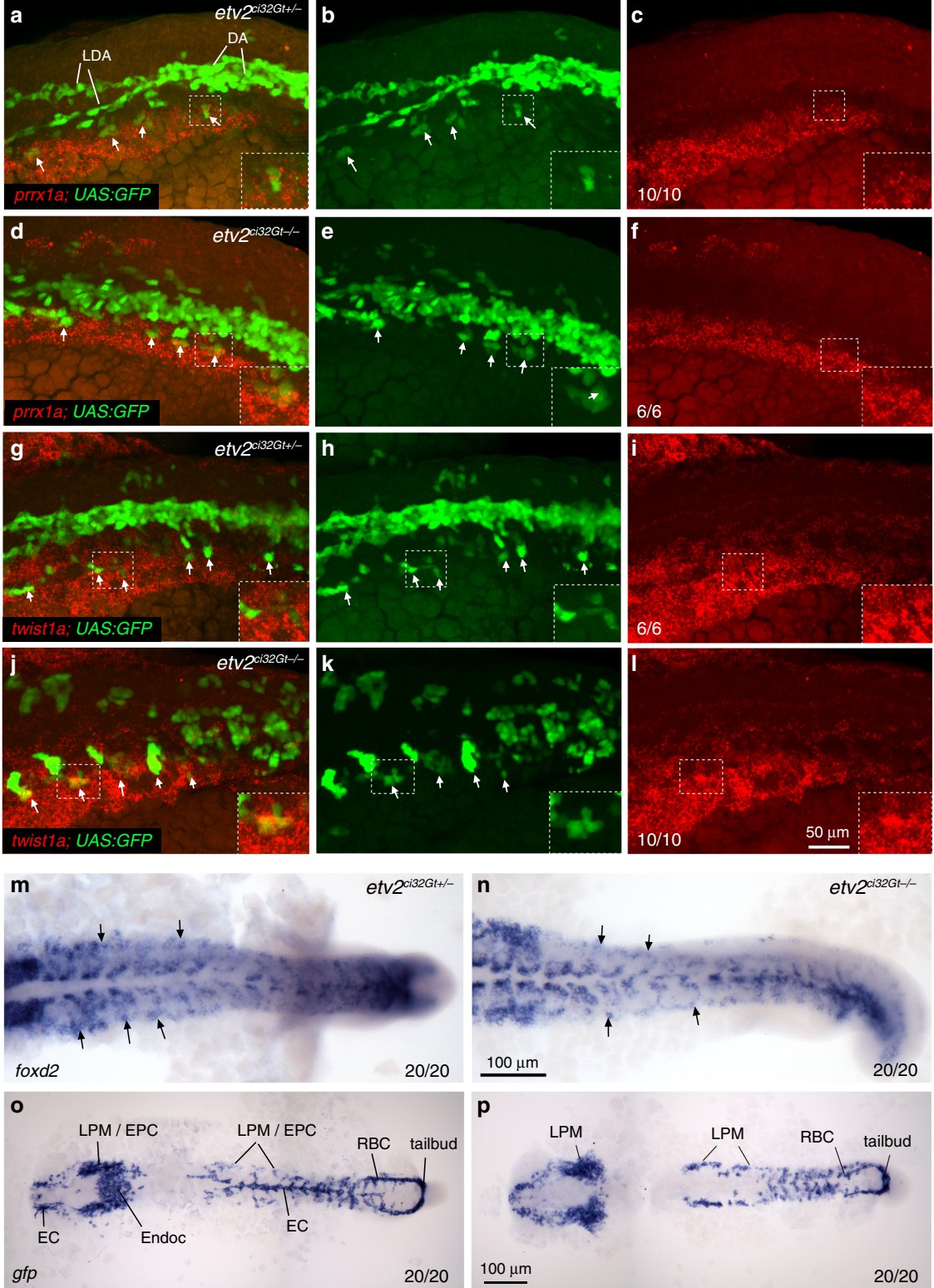

genes, including *id1* and *id3*. Loss of *msgn1/myod/myf5* function or activation of *hs:id3* expression resulted in a dramatic expansion of *etv2* expression into the paraxial mesoderm[40]. Our results show that a loss of *etv2* expression results in the opposite phenotype, and endothelial progenitors differentiate as skeletal muscle cells. FGF signaling was required for this differentiation, consistent with its role in promoting muscle fate. Similar loss of endothelial

cells and expansion of skeletal muscle was observed upon inhibition of BMP signaling using the *HS:dnBMPR* line[40], suggesting that BMP inhibits paraxial mesoderm development through *etv2* function.

Intriguingly, based on our scRNA-seq analysis, *id1* and *id3* genes were enriched not in the population of vascular progenitors, but in the LPM cells, together with *twist1a*, *prrx1a*, and

**Fig. 7 Expression of LPM cluster genes partially overlaps with *etv2^ci32Gt^; UAS:GFP* expression. a–l** Fluorescent in situ hybridization using hybridization chain reaction for *prrx1a* (**a–f**) and *twist1a* (**g–l**) expression combined with GFP fluorescence in *etv2^ci32Gt^; UAS:GFP* heterozygous or homozygous embryos at the 18-somite stage. Maximum intensity projections of selected confocal z-stacks are shown. Note that both *prrx1a* and *twist1a* are expressed bilaterally, and their expression partially overlaps with GFP in the most lateral cells (arrows). GFP+ cells fail to coalesce into axial vasculature in *etv2^ci32Gt−/−^* embryos. DA precursor vessel for the dorsal aorta, LDA lateral dorsal aortae. **m, n** Expression of *foxd2* in the trunk region in *etv2^ci32Gt^* heterozygous and homozygous embryos at the 18-somite stage. Note expression in the LPM region (arrows). Expression in the somites is also apparent. **o, p** In situ hybridization for GFP expression in *etv2^ci32Gt+/−^* and *etv2^ci32Gt−/−^; UAS:GFP* embryos at the 17–18-somite stage. GFP expression is observed in endothelial cell (EC, combined EC1 and EC2), endocardial (Endoc), endothelial progenitor cell (EPC), lateral plate mesoderm (LPM), tailbud and red blood cell (RBC) populations. Note the absence of EC and endocardial expression in (**p**). In all panels, the numbers display the number of embryos showing the expression pattern out of the total number of embryos analyzed in two replicate experiments.

*foxd1/d2* homologs. This suggests that BMP signaling is active in the multipotent progenitors in the LPM and activates *etv2* expression through *id1* and *id3*, which are then downregulated as cells differentiate into endothelial or hematopoietic lineages (Fig. 9). Based on our results, *etv2* mutant embryos showed a nearly complete absence of vascular endothelial and progenitor cells while the LPM population was increased, suggesting that many LPM cells are arrested in the progenitor stage unable to differentiate further, while others undergo differentiation into skeletal muscle.

A previous study has suggested that *id1* and *id3* genes function upstream of *etv2* during embryogenesis[40]. Here, we provide evidence that *prrx1a* and *foxd1/d2* are co-expressed with *etv2* in the LPM, and are involved in regulating its expression during vasculogenesis. This further supports the model that the LPM population includes early progenitors that can give rise to the vascular endothelial lineage and suggests that additional factors which function upstream of *etv2* may be present within this cell population.

While *etv2^ci32Gt−/−^* mutant embryos displayed UAS:GFP expression in multiple skeletal muscle cells, some muscle cells were also present in the heterozygous *etv2^ci32Gt+/−^* embryos. Because no GFP expression in skeletal muscle cells was observed in wild-type *Tg(-2.3etv2:GFP)* embryos, this phenotype is likely caused by the loss of a single *etv2* allele in heterozygous *etv2^ci32Gt+/−^* embryos. Intriguingly, vascular defects observed in *etv2^ci32Gt−/−^* mutant embryos were more severe compared to *etv2^y11^* and *etv2^ci33^* mutants, both of which are expected to be null alleles. A possible explanation is that the insertion of a rather large Gal4-containing construct into the *etv2* locus may have interrupted the expression of a related ETS transcription factor *fli1b*, which is positioned adjacent to *etv2* on the same chromosome and may share common regulatory elements with *etv2*. The combined loss of Etv2 and Fli1b function results in a similar (albeit more severe) loss of vascular endothelial differentiation as seen in homozygous *etv2^ci32Gt−/−^* embryos[49].

Overall, scRNA-seq of *etv2^ci32Gt^* and *Tg(-2.3 etv2:GFP)* embryos using the Chromium and Fluidigm platforms, respectively, identified some of the same cell populations, including endothelial progenitor cells, vascular endothelial cells, red blood cells, macrophages and presumptive tailbud progenitors. The cell populations which were observed in *etv2^ci32Gt^* heterozygous embryos but not in *Tg(-2.3 etv2:GFP)* embryos, include cardiomyocytes, skeletal muscle cells, apoptotic cells, and LPM progenitors. We have not observed any GFP expression in cardiomyocytes, skeletal muscle or apoptotic cells in *Tg(-2.3 etv2:GFP)* or *TgBAC(etv2:GFP)* embryos, therefore the presence of these populations is likely caused by a partial loss of *etv2* function in *etv2^ci32Gt+/−^* embryos. Indeed, loss of Etv2 function is known to result in EC apoptosis[21,49], and we have previously demonstrated that *etv2*-positive cells can differentiate into cardiomyocytes in *etv2*-inhibited embryos[8]. There are a couple of reasons as to why LPM progenitors were not identified by scRNA-seq of *Tg(-2.3 etv2:GFP)* embryos. It is possible that a

low number of LPM cells could not be separated into a distinct cluster because only 96 cells were sequenced using the Fluidigm C1 instrument. Also, this population may have low-GFP expression as these cells may have just initiated *etv2* expression, making them particularly difficult to detect by FACS. The *etv2^ci32Gt^; UAS:GFP* line is significantly brighter than the *Tg(-2.3 etv2:GFP)* line, likely due to the amplification of GFP expression by the Gal4:UAS system[14]. Therefore low GFP cells may have been missed when sorting cells from the *Tg(-2.3 etv2:GFP)* line.

The precise developmental timepoint at which vascular progenitors acquire either arterial or venous identities has been the subject of debate. Some experiments have suggested that arterial and venous identities are prespecified at very early stages during gastrulation or early somitogenesis[45,46]. In contrast, our results show that many arterial and venous genes are co-expressed in the same cells as late as the 20-somite stage, when arterial progenitors have begun assembling into vascular cords. These results support the model where vascular progenitors are initially bipotent and can differentiate as either arterial or venous cells. Intriguingly, venous progenitors which are known to originate more laterally and later in development than arterial progenitors[36], do not show significant expression of arterial markers. This suggests differences in the mechanisms involved in the early specification of arterial and venous progenitors.

In summary, our work suggests that cells in the vascular endothelial lineage arise from multipotent progenitors in the LPM which can differentiate into skeletal muscle in the absence of Etv2 function. The role of the key transcription factor Etv2 and the molecular mechanisms governing vascular development are highly conserved between many vertebrates, and therefore it is highly likely that our findings will also be relevant for mammalian embryos. Gaining insight into the fundamental mechanisms of cell fate commitment will not only provide us with a deeper understanding of complex developmental processes but will also be important for stem cell differentiation into highly diverse and specific lineages for regenerative therapies.

## Methods

**Zebrafish lines**. The following zebrafish lines were used in the study: *Tg(5xUAS: EGFP)*[50], *Tg(UAS-E1B:NTR-mCherry)*[51], *Tg(etv2:mCherry)^zf37^*[31], *Tg(-2.3 etv2:GFP) ^zf37^*[31], *TgBAC(etv2:GFP)^ci1^*[22], *etv2^y11^*[21], *Tg(fli1a:GFP)^y1^*[52], *Tg(gata1a:dsRed)*[53], *Tg(actc1b:GFP)*[32], *Tg(hsp70l:dkk1b-GFP)*[54], abbreviated further as *hsp70:dkk1*, *Tg (hsp70l:dnfgfr1a-EGFP)*[39], abbreviated as *hsp70:dnfgfr1a*. The *etv2^Gt(2A-Gal4)ci32^* gene trap line (further abbreviated as *etv2^ci32Gt^*) was generated by performing a knock-in of a construct with *gal4-pA* sequence into exon 5 of the *etv2* gene by CRISPR/Cas9 mediated homology-independent repair mechanism[14]. The *etv2^ci33^* mutant line was generated by injecting *Cas9* mRNA and *etv2* gRNA mixture (gRNA targets the following sequence within the exon 5 of *etv2*: GGGGAAAGGCCCAAGTCACA GAGG, PAM sequence is underlined) into embryos from the *kdrl:mCherry^ci5^* line[22]. The line contains a 13 bp insertion in the region targeted by the gRNA, designated in capital letters:…ggcccaagtcaGCTCTGCTGCCTGcagagg.

*Tg (5xUAS:loxP-mCerulean-pA-loxP-NTR-2A-YFP-pA)^ci46^* line (abbreviated as *UAS:mCer*) was made using Tol2-mediated transgenesis approach. The line showed mCerulean expression in all vascular endothelial cells in non-mosaic fashion when mated to *etv2^ci32Gt^* line.

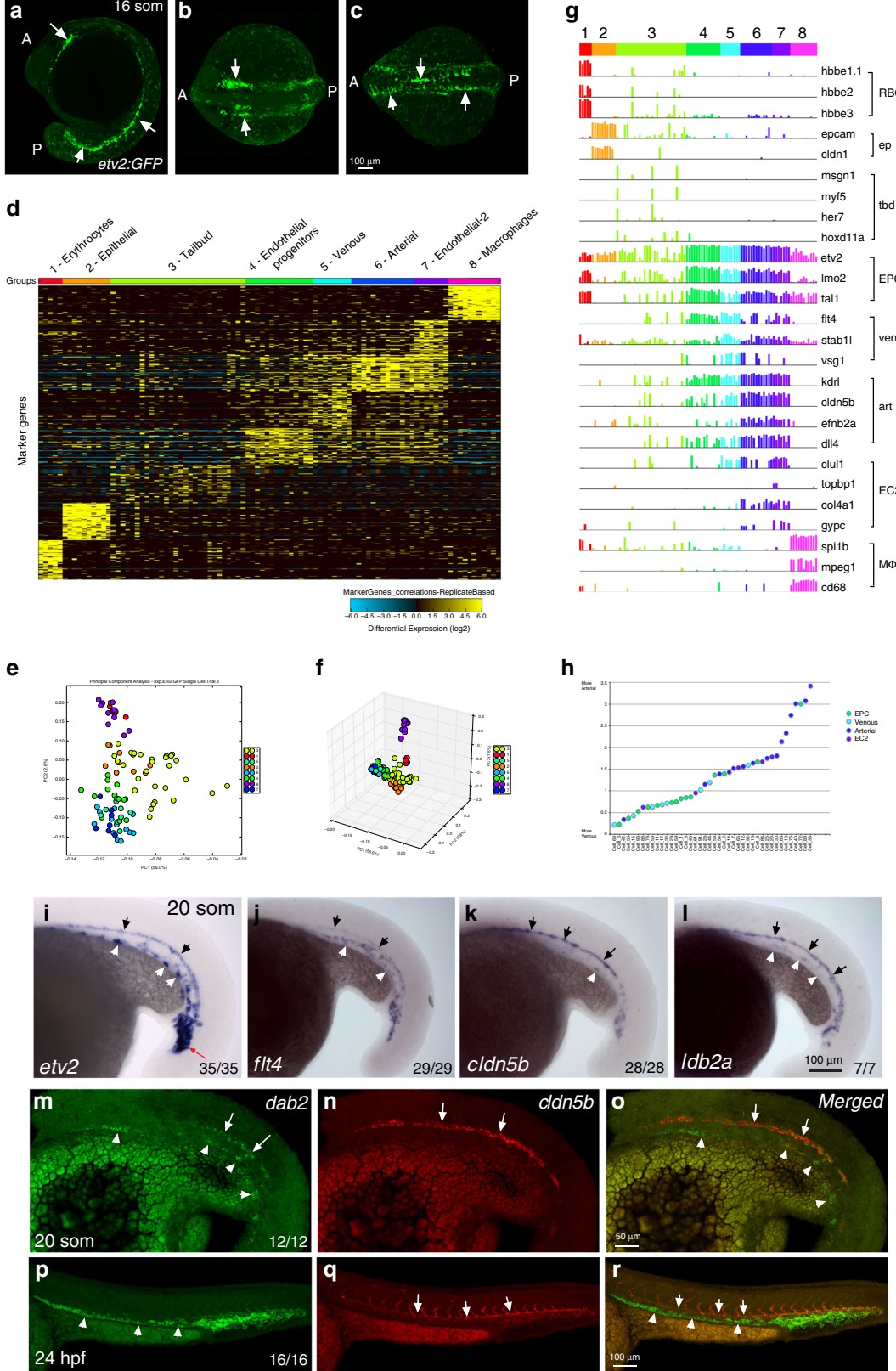

Throughout the study, embryos were typically incubated at 28.5 °C, except for the embryonic stages prior to 24 hpf, where embryos were incubated for part of the time at 23.5–24 °C to slow down their development. Embryos were staged using previously established criteria[55].

**MO injections**. Totally, 10 ng (per embryo) of previously validated *scl* splice-blocking MO was used for injections: AATGCTCTTACCATCGTTGATTTCA[56].

To knockdown *etv2*, 10 ng mixture (5 ng each MO) of two previously described *etv2 (etsrp)* MOs was used: MO1, TTGGTACATTTCCATATCTTAAAGT and MO2, CACTGAGTCCTTATTTCACTATATC (Gene Tools, Inc.)[5].

**Skeletal muscle cell counts**. To count GFP- or mCherry-positive cells within the skeletal muscle, embryos were mounted in 0.6% low-melting-point agarose and the trunk region was imaged using a Nikon A1 confocal microscope at the CCHMC

**Fig. 8 Single-cell RNA-seq analysis using Fluidigm cell sorting of *Tg(-2.3 etv2:GFP)* embryos at the 16–20-somite stage. a–c** GFP expression in live embryos, maximum intensity projection is shown. Arrows label vascular endothelial cells and their progenitors. Ten embryos were imaged in two independent experiments and a representative embryo is shown. **a** lateral view; **b** anterior view, **c** dorsal view. A anterior, P posterior. **d** Heatmap view of marker gene expression in different cell clusters. A complete list of differentially expressed genes is presented in Supplementary Data 5. **e, f** 2-D and 3-D principal component analysis plots of different cell clusters. Cluster names are the same as in (**d**). **g** Relative marker gene expression in different cell clusters. Vertical bars depict log-normalized gene expression. **h** An Arteriovenous (A-V) index of different endothelial cells. Note that many cells are positive for both arterial and venous marker expression. **i–l** ISH expression analysis of key marker genes for EPC (*etv2*), venous (*flt4*), arterial (*cldn5b*), and EC-2 (*ldb2a*) populations in the trunk region at the 20-somite stage. Black arrows label the DA and white arrowheads label venous progenitors which are starting to coalesce into the PCV. Note that *flt4* is enriched in the PCV while *cldn5b* and *ldb2a* label the DA. **m–r** Two color ISH analysis for the expression of venous *dab2* and arterial *cldn5b* at the 20-somite and 24 hpf stages. Arrows label the DA while arrowheads mark the PCV or its progenitors. Note that *dab2* and *cldn5b* are co-expressed in the DA progenitors at the 20-somite stage but not at 24 hpf. In all panels, the numbers in the lower right corner display the number of embryos showing the expression pattern out of the total number of embryos analyzed in two replicate experiments.

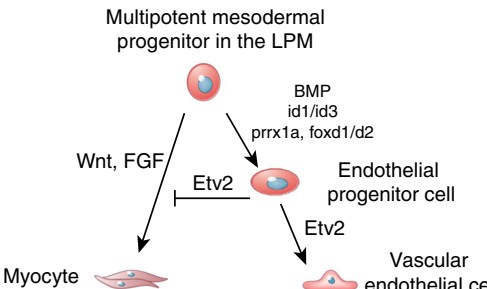

**Fig. 9 A proposed model for the differentiation of vascular endothelial cells from the multipotent mesodermal progenitors in the LPM.** Wnt and FGF signaling promotes myocyte differentiation of multipotent progenitors in the lateral plate mesoderm (LPM). BMP signaling through its downstream effectors *id1* and *id3* promotes vascular endothelial differentiation. Additional LPM markers that include *prrx1a, foxd1, foxd2* and others may be involved in this process, although their role is purely speculative at this point. Etv2 promotes vascular endothelial differentiation while directly or indirectly repressing myogenesis.

Confocal Imaging Core. Fluorescent cells within the skeletal muscle were counted manually within the entire trunk region by analyzing confocal z-stacks.

**In situ hybridization**. To perform ISH, embryos were fixed overnight in BT-Fix (4% paraformaldehyde in 1× phosphate-buffered saline (PBS)), dehydrated in sequential ethanol series and stored at −20 °C. Embryos were rehydrated and washed 3× in PBT (1× PBS, 0.2% bovine serum albumin, 0.2% Tween 20). Embryos were incubated in the prehybridization buffer (50% formamide, 5× SSC, saline sodium citrate, 50 µg/ml heparin, 5 mM EDTA, 0.5 mg/ml rRNA, 0.1% Tween 20, citric acid added to pH 6.0) for 2 h at 65 °C. DIG-labeled RNA probe solution was added at approximately 1 µg/ml concentration in prehybridization buffer (prehyb) and incubated overnight at 65 °C with shaking. Embryos were washed at 65 °C with subsequent solutions: 75% prehyb/25% 2× SSC; 50% prehyb, 50% 2× SSC; 25% prehyb, 75% 2× SSC; 2× SSC; twice with 0.2× SSC. Subsequently, embryos were washed at room temperature in the following solutions: 75% 0.2× SSC, 25% PBT; 50% 0.2× SSC, 50% PBT, 25% 0.2× SSC, 75% PBT, and 100% PBT. Afterwards embryos were incubated overnight in 1× PBT with 2% lamb serum and 1:4000 dilution of anti-digoxigenin-AP antibody (Sigma-Aldrich, cat No. 11093274910). Embryos were washed 6× in PBT solution and incubated in AP buffer (100 mM NaCl, 50 mM MgCl₂, 100 mM Tris-Cl, pH 9.5, 0.1% Tween 20) in the presence of 4-nitrotetrazolium blue chloride (0.225 mg/ml) and 5-Bromo-4-chloro-3-indolyl phosphate disodium salt (0.175 mg/ml). Staining was stopped with PBT washes.

The following probes were used: *etv2*[57], *cldn5b*[58], *flt4*[18], *fn1*[59], *drl*[60]. To synthesize a *gfp* probe, a *gfp* fragment was amplified by PCR from the *myl7-hand2-IRES-GFP* vector[61] using the following primers: ATGGTGAGCAAGGGCGAGG AG and ATTATGCTGAGTGATATCCCTTACTTGTACAGCTCGTCC followed by RNA synthesis using T7 RNA polymerase (Promega) and DIG-labeling mix (Sigma-Aldrich). To synthesize a *ldb2a* probe, a cDNA clone from GE Dharmacon, cat. No. MDR1734-202728743, which contains *ldb2a* cDNA in pME18S-FL3 vector, was used as a template for PCR amplification with primers 18S-1 (CTTC TGCTCTAAAAGCTGCG) and 18S-T7 (CCTTTAATACGACTCACTATAGGG CCGCGACCTGCAGCTCG). DIG-labeled RNA was transcribed from the PCR product using T7 RNA polymerase (Promega). A *foxd2* cDNA fragment was amplified by PCR from zebrafish embryonic cDNA using primers TCGGACAGTT CTGCTCTGTC (Forward) and TAATACGACTCACTATAGGGCTTGCTTCGG

CCACGAACCA (Reverse with T7 sequence), followed by RNA synthesis using T7 RNA polymerase. T3 and T7 primers were used for PCR amplification of *lmo2* cDNA in pBK-CMV vector[18], followed by in vitro probe synthesis using T7 RNA polymerase.

Fluorescent ISH for *prrx1a*, *twist1a*, *dab2*, *cldn5b*, and *myod* expression was performed using hybridization chain reaction (HCR, version 3)[62]. HCR probes and fluorescent hairpins were synthesized by Molecular Technologies at Beckman Institute, California Institute of Technology, Pasadena, CA. Embryos were fixed, dehydrated and rehydrated using standard ISH protocol as described above. Embryos were prehybridized in hybridization buffer (30% formamide, 5× SSC, 9 mM citric acid (pH 6.0), 0.1% Tween 20, 50 µg/ml heparin, 1× Denhardt's solution, 10% dextran sulfate) for 30 min at 37 °C. Two picomole of each HCR probe was added to 50 µl of hybridization buffer and incubated overnight at 37 °C. Embryos were washed 4 × 15 min in 30% formamide, 5× SSC, 9 mM citric acid (pH 6.0), 0.1% Tween 20, 50 µg/ml heparin solution at 37 °C. Samples were then washed 3 × 5 min with 5× SSCT (5× SSC and 0.1% Tween 20) at room temperature, followed by 30 min incubation in amplification buffer (5× SSC, 0.1% Tween 20, 10% dextran sulfate) at room temperature. Thirty picomole of each fluorescently labeled hairpin by snap cooling 10 µl of 3 µM stock solution. Hairpin solutions were added to embryo samples in amplification buffer and incubated overnight in the dark at room temperature. Samples were washed 5× with 5× SSCT solution, and stored at 4 °C until imaging. GFP fluorescence was still apparent after HCR. Embryos were mounted in 0.6% low-melting point agarose and imaged using a Nikon A1 confocal microscope at the CCHMC Confocal Imaging Core.

**Immunostaining**. F310 (1:50 dilution) and S58 (1:10) primary antibodies (Developmental Studies Hybridoma Bank, Iowa City, IA) were used for whole-mount immunostaining of *etv2*[ci32Gt]; *UAS:GFP* embryos at 24 hpf. No immunostaining was performed for GFP which was visible in fixed embryos. Anti-mouse IgG, CF594 antibody (Sigma-Aldrich, SAB4600098) was used as a secondary antibody.

**Microscopy imaging and image processing**. Embryos were whole mounted in slide chambers with 0.6% low-melting poing agarose. To image stained embryos after ISH, images were captured using a 10× objective on an AxioImager Z1 (Zeiss) compound microscope with an Axiocam ICC3 color camera (Zeiss). Images in multiple focal plans were captured individually and combined using the Extended Focus module within Axiovision software (Zeiss). For confocal imaging, embryos were mounted in 0.6% low-melting point agarose and imaged using a 10×, 20×, or 40× objective on a Nikon A1R confocal microscope. Denoising (Nikon Elements software) was performed for some of the images with weak signal to reduce noise. Images were assembled in Adobe Photoshop CS6 software package. Non-linear level adjustment was used to increase contrast and reduce background. In all cases, images of control and experimental embryos were adjusted similarly.

***etv2* overexpression and quantitative RT-PCR**. *etv2* mRNA was synthesized as previously described[5]. To analyze expression of muscle markers, approximately 75 pg of zebrafish *etv2* mRNA was injected into *fli1a:GFP* embryos at the 1-cell stage. Embryos were screened for ectopic GFP expression at the 10-somite stage and then frozen for qPCR analysis. Groups of 10 embryos were analyzed in two independent experiments. An RNAqueous 4-PCR kit (ThermoFisher) was used to extract RNA. cDNA synthesis was performed using Superscript IV cDNA synthesis kit (ThermoFisher). Quantitative real-time PCR was performed using SYBR Green Master Mix (ThermoFisher) and StepOne Software v2.3 (Applied Biosystems). The following primers were used:

*ef1α* (TCACCCTGGGAGTGAAACAGC) and (ACTTGCAGGCGATGTGAG CAG),

*myf5* (GGTTGACTGCAACAGTCCTG) and (GCGTTGGCCTGAGGCATC TT),

*myog* (GCATAACGGGAACAGAGGCA) and (CAGCCTTCCTGACTGCCT TA),

*myod* (CCAGCATCGTGGTGGAGCGAATT) and (GGTCGGATTCGCCTTT TTCT).

The results were analyzed using StepOne Software v2.3 (Applied Biosystems). Two biological and two technical replicates were obtained for each sample (four biological replicates were available for control uninjected embryos). Relative expression values were normalized for *ef1α* expression. Statistical analysis was performed using Prism 8 software (GraphPad Software).

**Single-cell RNA sequencing using the Chromium platform.** Approximately, 100–150 of *etv2ci32Gt+/−; UAS:GFP* and 75–100 *etv2ci32Gt−/−; UAS:GFP* embryos were collected at the 20-somite stage and dissociated into a single-cell suspension using the previously reported protocol[63]. Briefly, the embryos were manually dechorionated, and transferred into the deyolking buffer (55 mM NaCl, 1.8 mM KCl, and 1.25 mM NaHCO₃). Embryos were pipetted up and down on ice using p1000 pipettor until yolk was dissolved. Embryos were centrifuged at 300 G for 1 min, and supernatant was removed, while the pellet was resuspended in 0.5× Danieau solution (1× Danieau: 58 mM NaCl, 0.7 mM KCl, 0.4 mM MgSO₄, 0.6 mM Ca(NO₃)₂, 5 mM HEPES, pH 7.6). Centrifugation and removal of supernatant was repeated again, followed by another centifugation at 300 G. The pellet was resuspended in FACSmax solution cell dissociation solution, which was then passed through a cell strainer. Cells were centrifuged at 300 G and suspended in a buffer of 1× PBS containing 1 mM EDTA and 2% fetal bovine serum (FBS). Fluorescence-activated cell sorting was then used to collect GFP+ cells in a solution of 50% FBS in 1× PBS (see Supplementary Fig. 16a for FACS gating strategy). Single cells were captured and processed for RNA-seq using the Chromium instrument (10× Genomics) at the CCHMC Gene Expression Core facility. RNA-seq was performed at the CCHMC DNA Sequencing core on Illumina HiSeq2500 sequencer using one flow cell of paired-end 75 bp reads, generating 240–300 million total reads.

Cell Ranger version 2.0.0 was utilized for processing and de-multiplexing raw sequencing data. Raw basecall files were first converted to the fastq format, and subsequently the sequences were mapped to the Danio rerio genome (version zv10) to generate single-cell feature counts (using STAR alignment). Downstream analysis of the gene count matrix generated by CellRanger was performed in R version 3.5.2 using Seurat version 2.3.4[64,65] and Tidyverse packages. The gene counts matrix was loaded into Seurat and created by filtering cells which only expressed more than 200 genes and genes that were expressed in at least 3 cells. In addition, as an extra quality-control step, cells were filtered out (excluded) based on the following criteria: <200 or >3500 unique genes expressed, or >5% of counts mapping to the mitochondrial genome. This resulted in 2049 cells in the *etv2ci32Gt* heterozygous sample and 588 cells in the *etv2ci32Gt* homozygous sample. For further downstream analysis, data from the heterozygous and homozygous samples were merged, resulting in a combined dataset of 2637 cells. Raw read counts were normalized by the "LogNormalize" function that normalizes gene expression levels for each cell by the total expression, multiplies the value by a scale factor of $10^4$ and then log-transforms the result. Highly variable genes were determined by calculating the average expression and dispersion for each gene, placing the genes into bins and then calculating a *z*-score for dispersion within each bin. Based on the parameters selected to remove outliers, 3707 genes were calculated as being highly variable in expression and these genes were used for downstream analysis. Prior to dimensionality reduction, a linear transformation was performed on the normalized data. Unwanted cell-cell variation driven by mitochondrial gene expression and the number of detected molecules (nUMIs) was removed by regressing out these variables during the scaling of data.

Dimensionality reduction was performed on the entire dataset using principal component analysis using the list of highly variable genes generated above. The top 13 principal components which explained more variability (than expected by chance) were identified based on PC heatmaps, the JackStrawPlot and PCElbowPlot. These 13 components were used as input for generating clusters (using the default SLM algorithm), with a resolution of 0.6. t-SNE[16] was utilized to reduce the dimensionality of the data to two dimensions (for visualization purposes). Following clustering, genes differentially expressed in each of the clusters were determined using a method of differential expression analysis based on the non-parametric Wilcoxon rank sum test. Cells in each cluster were compared against cells of all other clusters. Genes were then filtered based on being detected in ≥25% of cells within a cluster and a Bonferroni adjusted *p* value < 0.05. Based on the lists of differentially expressed genes ordered by average log fold change, clusters were assigned specific cell identities. Visualization of specific gene expression patterns across groups on t-SNE and violin plots was performed using functions within the Seurat package.

To subcluster endothelial cells, prior to merging data from *etv2ci32Gt* heterozygous and homozygous cells, clustering was performed on the two datasets independent of each other. For the *etv2* heterozygous dataset, 15 significant principal components were identified and used as input for t-SNE dimensionality reduction, using a resolution of 1.2 for clustering. An endothelial cell population was identified based on differentially expressed genes in that cluster. To identify heterogeneity within the endothelial cell population, the endothelial cells were subsetted into a separate Seurat object and highly variable genes were calculated. A linear transformation was performed again whilst removing unwanted variation driven by mitochondrial gene expression and nUMI by regressing out these variables during the scaling of data. Fourteen significant principal components were selected for t-SNE dimensionality reduction, using a resolution of 1.0 for clustering. Based on these parameters, there appeared to be two transcriptionally distinct populations of endothelial cells, which was also confirmed by a heatmap of gene expression of the two groups.

**Pseudotime analysis.** To analyze endothelial cell lineage pathways, we used Monocle version 2.8.0 that implements reversed graph embedding on gene expression data to determine single-cell developmental trajectories in an entirely unsupervised manner[66]. The following populations of cells (from merged heterozygous and homozygous data) were subsetted and imported to Monocle: LPM, EPCs, EC1, EC2, Myocytes. Cells with a UMI count > 10⁶ were excluded from the data to exclude potential multiplets. Furthermore, upper and lower bounds on nUMI were set at two standard deviations above and below the mean UMI to remove low-quality cells. Only cells falling within these boundaries were included for further downstream analysis Filtering of genes was performed by keeping genes expressed in ≥10 cells. Gene dispersion values and average expression was calculated and genes with a mean expression ≥0.1 were subsetted. The top seven significant principal components were used to reduce the dimensionality of the data using t-SNE. Cells were then clustered, setting num_clusters = 8. A differential gene test was performed to determine significant differentially expressed genes and the top 425 genes were arranged by *q* value to create a list of ordering genes. An ordering filter was then applied to the cells based on the list of genes created in the previous step to label whether a cell is used for ordering or not. In order to determine cell differentiation trajectories, dimensionality reduction to two components was performed using the "DDRTree" method and then assignment of pseudotime and state to each of the cells was carried out using the OrderCells function. The trajectories of cells were plotted on a pseudotime graph and colored by cell identities previously determined using clustering in Seurat.

**Single-cell RNA sequencing using Fluidigm C1.** Sequencing was performed on cells from zebrafish embryos at 16–18 somite stages and at 20 somite stages. Approximately, 300 *Tg(-2.3 etv2:GFP)* embryos were collected at each stage, dissociated into a single-cell suspension as described above and suspended in a buffer of 1× PBS containing 1 mM EDTA and 2% FBS. Fluorescence-activated cell sorting was then used to collect GFP+ cells in a solution of 50% FBS in 1× PBS (Supplementary Fig. 16b). Cells were then counted on a hemocytometer, spun down at $300 \times G$ and resuspended at a concentration of approximately 30,000 cells per ml. Each suspension was then loaded onto a 96-well Fluidigm C1 chip for capture of single cells at the CCHMC Gene Expression core. Forty-eight individual cells from each stage group were selected for library synthesis, barcoding, and pooled paired-end sequencing using a single lane of a PE75 flow cell at 1.5–2 million reads per cell, performed at the CCHMC DNA Sequencing core. Data analysis were performed using AltAnalyze[44]. Reads were aligned to Ensembl release 72 of the Zv9 assembly of the zebrafish genome and gene counts were quantified using Kallisto[67]. Iterative Clustering and Guide-gene Selection (ICGS) was used to identity de novo clusters/groups of cell[68]. ICGS was performed using cosine similarity with an expression fold cutoff of six for a minimum of five cells using conservative filtering for cell cycle effects. From ICGS, group specific enriched genes were identified automatically using Markerfinder[69]. All heatmaps and principal component analyses were created using AltAnalyze.

For the enriched marker table for each cluster, log-normalized fold change values were acquired directly from the AltAnalyze MarkerFinder analysis. Averages of log-normalized fold-change were generated for each marker per cluster. P-value was determined by a two-tailed Mann–Whitney *U* test comparing fold change in the cluster of interest to fold change in all other cells. The table of average expression values for genes in each cluster was created directly from a text file generated from AltAnalyze. These values represent the average of expression of all cells in that cluster calculated as reads for a specific gene that have been normalized to the total reads per cell and a scale factor of 10,000× applied.

To obtain arteriovenous index, the average of the log₂ expression was calculated first for the following known arterial and venous markers: *cldn5b, efnb2a, dll4, hey2, dlc, notch*3 (arterial), and *flt4, dab2, stab1l, ephb4a, mrc1a* (venous). Subsequently, the average arterial expression was divided by the average venous expression to generate an arteriovenous identity index (AV index).

**Fluorescent cell counting.** Percentage of fluorescent cells which express *etv2* reporter was estimated in *etv2ci32Gt+/−; UAS:mCerulean (mCer)* embryos alone or crossed with *hsp70:Dkk1-GFP* and *hsp70:dnFGFR1-GFP* lines. Cells from 10 to 25 embryos in each group were disaggregated at 24 hpf as described above and mCer-positive cells were counted using BD FACS Aria II at the CCHMC FACS core facility.

To count *gata1:dsRed*-positive cells in wild-type and *etv2* MO-injected embryos, cells from 50 to 70 embryos in each group were disaggregated at approximately 23 hpf. dsRed-positive cells were counted at the CCHMC FACS core facility using BD FACS Canto II. FACS gating strategy for both fluorescent lines is shown in Supplementary Fig. 16c, d.

**Ethics approval**. Zebrafish embryo experiments were performed under animal protocol. IACUC2016-0039, approved by the Institutional Animal Care and Use Committee at the. Cincinnati Children's Hospital Medical Center.

**Reporting summary**. Further information on research design is available in the Nature Research Reporting Summary linked to this article.

## Data availability

All data that support the findings of this study are available from the corresponding author upon reasonable request. scRNA-seq data generated in the study using the Chromium (10× Genomics) platform have been deposited in the NCBI GEO database under the accession code GSE143750. scRNA-seq data generated in the study using the Fluidigm platform have been deposited in the NCBI GEO database under the accession code GSE142484.

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

## Acknowledgements

This research was supported by the awards from National Insitute of Health: NIH R01 HL134815 and R21 AI128445 to S.S., and NIH F31 HL135986 to A.L.K. We thank Nathan Salomonis and Praneet Chaturvedi for their assistance with scRNA-seq analysis, Steve Potter for critical comments, and CCHMC Gene expression and FACS cores for their assistance with the project.

## Author contributions

B.C. generated etv2$^{ci32Gt}$ line and performed most of experiments in the study, S.C.C. performed scRNA-seq analysis of the Chromium data and edited the paper, A.L.K. performed all experiments using the Fluidigm platform and subsequent data analysis, as well as the cell disaggregation and initial data processing using the Chromium platform, S.S. initiated and supervised the entire study, contributed to data analysis, and wrote the paper.

## Competing interests

The authors declare no competing interests.
