## [Peer Review File · Nature Communications]

Reviewers' Comments:

Reviewer #1:

Remarks to the Author:

This manuscript by Chestnut and colleagues explores the consequences of ETV2 deficiency during zebrafish embryonic development. Using single cell RNA sequencing, they identify key missing populations and overall changes upon ETV2 deletion. Additionally, they define skeletal muscle as an alternative fate for ETV2-deleted progenitors. While this manuscript presents several novel and interesting data, the claims made by the authors would need to be supported by additional experimental evidences to be fully convincing. Finally, the last set of data on transcriptional profiling of ETV2:GFP cells and arterial versus venous fate is interesting in its own but distracting from the main message of the manuscript.

Specific comments:

Why was the *Etv2:gal4, UAS:GFP* rather than the *Etv2:GFP* or *Etv2:mCherry* used for the scRNA-seq experiments? Why not do an ETV2-2A-GFP genomic insertion? Is there an advantage in using the *gal4/UAS* transgenic system? How accurate and stringent is *Etv2* expression represented by GFP in this model? In addition to the immunofluorescent data presented, was any molecular analysis performed? What is the half-life of GFP in this transgenic model? As judged by *etv2* expression in single cell RNA-seq data (fig 1 c), only a low fraction of the cells analyzed do express *etv2*.

In the scRNA-seq experiment, 2049 GFP+ cells were sequenced from 100-150 het embryos and 588 GFP+ cells from 75-100 from homo embryos. It is not clear how representative the scRNA-seq data are of overall GFP+ embryonic populations. What are the frequencies of GFP+ cells in het and homo embryos? This becomes very important when reporting the proportions of identified subsets as shown in figure 1e for example. The authors state that: "the LPM, cardiomyocyte and skeletal muscle (myocyte) populations were greatly increased in the homozygous embryos" but is this increase only relative within the context of the single cell populations analysed? It is not clear that these conclusions can be extrapolated to the full embryo.

Page 8 lines 40-41: "The presence of GFP-positive cardiomyocytes in *etv2:gal4* heterozygous embryos could be explained by the loss of one functional *etv2* allele, which resulted in some *etv2+* cells differentiating as cardiomyocytes." What are the evidences that *Etv2+* progenitors do not give rise to cardiomyocytes? Has lineage tracing with *Etv2* been performed?

Page 3, lines 38-39: "*Etv2* function is required to actively repress alternative cell fates in multipotent mesodermal progenitors." What are the evidences that these progenitors are multipotent and that *Etv2* is expressed in these progenitors? Based on the scRNA-seq data, few LPM cells express *etv2*. If those represent progenitors, one might expected to find *etv2* still expressed.

Page 9 lines 5-6: "A population of RBC cells was expanded in the *etv2:gal4* homozygous embryos". What is the basis for claiming that RBC cells expanded? Are there more RBCs in *Etv2:gal4+/+* embryos? This should be supported by additional evidences.

Differentiation of *etv2+* progenitors into skeletal muscle: it is not clear how prevalent this is? What is the frequency of *etv2+* derived skeletal muscle relative to the overall number of skeletal muscle cells at the embryonic stage analyzed (figure 3, 4)? Is this a marginal event or a widespread phenomenon? The *scl* morpholino data are rather weak and would need additional molecular-based evidences to support the involvement of *Scl* in this process.

Page 10, lines 31-32: "Muscle-specific myod expression was greatly inhibited upon *etv2* overexpression". This does not strengthen the conclusion given that it is well established that *etv2*

overexpression in many cell types will push cells toward a vascular fate. Lack of myod expression only reflect a lack of skeletal muscle cells but does not directly implicate etv2 as a repressor of skeletal muscle fate.

In the dkk1 treated embryos (fig 4), what happens globally to myocytes? Are they reduced to the same extend as the GFP+ myocytes? The authors state that "Wnt signaling promotes muscle differentiation in multipotent etv2+ progenitors." Is this unique to the etv2+ derived myocytes?

Page 14, lines 43-44: "our work has demonstrated that cells in the vascular endothelial lineage arise from multipotent progenitors in the lateral plate mesoderm which can differentiate into skeletal muscle in the absence of Etv2 function". To fully demonstrate this, lineage tracing would be required.

Minor comments

Page 10, lines 10-11: " Double heterozygous etv2gal4/ci33 embryos displayed a dramatic increase in GFP+ myocytes compared to etv2:gal4+/- embryos". It looks like a 3 to 4 fold increase in GFP+ myocytes which is far from "dramatic". Same remark for the sentence: "SU5402-treated embryos showed a dramatic decrease in the number of ectopic GFP+ muscle cells", data in which GFP+ myocytes decrease about two-fold."

Page 10, lines 23-24: "The number of myocytes was significantly increased in the etv2:gal4+/- embryos injected with the previously validated scl morpholino (MO)". Does this sentence refer to the overall number of myocytes per embryos?

Figure 5: it is very difficult to see co-expressing cells. Close-up images should be shown.

Page 14, lines 10-11: "previous studies have demonstrated the existence of a common vascular and endothelial progenitor, the hemangioblast". Is that not rather a common hematopoietic and endothelial progenitor?

Reviewer #2:

Remarks to the Author:

In this manuscript Chestnut et al., perform scRNA-seq in etv2-expressing cells from zebrafish embryos. They perform a thorough analysis and based on the expression of known markers for specific lineages they identify clusters of endothelial cells, myeloid and erythroid cells, LPM progenitors etc. Interestingly, they observe that some of the Etv2 expressing cells differentiate into skeletal muscle cells in the absence of Etv2. They then show that this fate is promoted by the Wnt and FGF pathways and assume that Etv2 is required to suppress this alternative fate. Apart from this main observation of the paper, they also identify the transcriptional signature of multipotent progenitors in LPM and show that arterial progenitors co-express arterial and venous markers. Single cell RNA seq technology gave a tremendous boost to developmental and lineage tracing studies and this paper provides important information on the molecular fingerprint of vasculogenesis. However, many aspects need to be addressed or clarified.

Major comments:

1. The authors need to find a way to isolate the etv2 positive cells and they employ a transgenic line that has gal4 inserted into the coding part of the gene. As a result, they use the heterozygous and homozygous cells of this line and treat the heterozygous as almost wild type. This can lead to wrong conclusions, especially if the authors don't show in detail that heterozygosity of etv2 does not disturb main functions. Another way to solve the problem would be to use the Tg(-2.3 etv2:GFP) line, perform scRNA-seq and compare it to their current results. It is understandable that these two lines do not mark exactly the same cell populations, but some commonalities

should be present. In addition, the authors claim that the phenotype of their transgenic animals is more severe than the one of the actual mutants. This may be caused by deletion of regulatory elements that are located close to *etv2* gene. Can the authors check the expression of these adjacent genes? Such a problem leads to an even more urgent need to sequence cells from the Tg(-2.3 *etv2*:GFP) or any other transgenic that could verify the current results.

2. The number of heterozygous and homozygous cells sequenced differs greatly between them. This maybe a problem in comparing these two experiments and identifying missing or enriched populations. Can the authors equilibrate these differences or prove that they do not affect the results?

3. The authors claim that Wnt and FGF signaling promote skeletal muscle fate. Can they show in their scRNA seq data that genes regulated by these pathways have altered expression?

4. Can *scl* overexpression rescue the phenotype?

Minor comments

1. Page 7 line 30: Supp. Fig. S2 should be mentioned here.

2. On Figure 1 the authors claim that they show also markers of erythroid cells and other populations (specifically in Fig 1c,d) but that is not so. Can the authors add all the markers that are mentioned in the text?

3. The authors claim that the homozygous transgenic animals have more erythroid cells. Can the authors verify this using an independent method like WISH or by FACS analysis of an erythroid reporter animal?

4. The authors show results on heterozygous and homozygous animals, but never on real wild type animals. Can the authors include wild type animals in any case that is pertinent, for example in WISH experiments?

5. Page 8 Line 16,17. Apoptosis and cell cycle genes do not exist in Fig 1c,d.

6. Page 10 line 2 *myod* expression is missing from the figure.

7. Can the authors clarify how they count the GFP+ myocytes?

Reviewer #3:

Remarks to the Author:

The authors have made a zebrafish transgenic line to characterise the development of cells derived from *etv2*-expressing progenitors by inserting a Gal4 protein into the *etv2* locus. They then isolated GFP+ cells from wildtype and *etv2* heterozygote animals at two stages of development when haematopoietic and vascular progenitor cells are specified and performed single cell sequencing. They used outputs from this sequencing to highlight a population of cells that express genes expressed in myogenic lineages and ask whether this reveals that *etv2*-expressing cells can form muscle. They test this by performing time-lapsed analysis of the same transgenic line to argue that a population of lateral-plate mesoderm (LPM) derived cells form myofibres during development and that this is prevented by expression of *etv2* in these cells. They then argue that *etv2* acts to promote a fate decision of LPM cells to form blood vessel cells and in an absence of *etv2* function there is a fate switch to myoblasts. They support this proposed mechanism by showing that knockdown of the *etv2* target gene *scl* increased the number of GFP+ myoblasts that form. They also show that over-expression of *etv2* RNA appears to reduce expression of *myod* at early stages.

They then investigate the importance of Fgf and Wnt signalling in directing this fate decision using transgenic lines to inhibit Wnt and Fgf and a small molecule inhibitor of Fgf receptor function. Based on changes to the number of GFP+ cells forming myofibres they argue that both Wnt and Fgf promotes *etv2*-expressing cells to assume a muscle fate.

The central premise of this work is that *etv2*-expressing cells in the LPM make a fate decision controlled by *etv2*/*scl* function relative to Wnt and Fgf activity. An important consideration when making this assumption is whether there is an effect on progenitor cells in the LPM that may affect how they respond to signals from their environment. It was not possible to discern from the time-

lapse movies how the GFP+ progenitor cells migrate in *etv2* heterozygotes and whether this is altered. If they show a different spatial organisation prior to or during migration they are likely to experience different cues that affect their specification. In order to address this point the authors should carefully characterise *etv2*-expressing cells in heterozygotes and homozygous animals using dorsal and lateral views to highlight cell movement to answer whether cells migrate differently in homozygous animals. A similar point holds for animals in which Wnt and Fgf are altered.

I was also not clear if there are changes to the relative number of GFP+ cells at different stages of development in both genotypes or in when Fgf or Wnt are inhibited as this may also explain why there are more or fewer GFP+ myofibres. Careful quantification of cell number relative to their position is needed to address this issue.

Given that the entire paper relies on the new *etv2* line I was surprised that this is not described in full in this paper. It is important to do this as I had the impression that the expression differs from a previously published line made with a BAC and it is not clear in this work how accurately the new CRISPR-generated line expresses GFP relative to *etv2*.

A similar caveat holds for descriptions for cell behaviour - it is simply not possible to see the different cell populations described in the text from the figures shown, specifically how the LPM progenitors migrate. Clearer figures demonstrating the spatial distribution of GFP+ cells is needed to clarify how this may affect their fate switch in an absence of *etv2* function.

The hypothesis that *etv2* acts to inhibit myogenesis is tested by injecting RNA for *etv2* and assaying *myod* expression. A striking result is shown indicating a unilateral loss of *myod* expression at early somite stages. This should be corroborated by qPCR for *myod* and other myogenic genes including *myf5*.

The statement that GFP+ cells are forming myofibres should be tested by antibody staining to show that these express markers of differentiated myofibres.

One important piece of evidence for the role of *etv2* in dictating fate of this early LPM progenitor population is to show where progenitor cells switch to a myogenic fate and when this occurs. This could be accomplished by examining *myod* expression relative to GFP+ cells in *etv2* mutants at different stages of development to show if there is ectopic expression in GFP+ cells and when this occurs.

Reviewer #4:

Remarks to the Author:

Chestnut B. et al performed a comprehensive study for *Etv2* deficient zebrafish embryos during hematovascular development. It is known that *Etv2* expression alone is sufficient to transdifferentiate skeletal muscle cells into functional blood vessels (Ref: Transdifferentiation of fast skeletal muscle into functional endothelium in vivo by transcription factor *Etv2*. *PLoS biology*, 11(6), e1001590.). The major finding of this study is that in the absence of *Etv2* function, vascular progenitors can acquire a skeletal muscle fate. Hence, this manuscript further confirmed the role of *Etv2* in blood vessel development.

Major comments:

(1) Over-expression of *Etv2* have been extensively studied. It will be interesting to compare this study (*Etv2*-deficient) with published *Etv2*-overexpression datasets, and see the overlap (e.g., whether key upregulation markers in this study are downregulated in *Etv2*-overexpression datasets).

(2) Page 5 (Line 17-19): To perform QC, it is reasonable to remove cells with low number of genes expressed (<200). But it is unclear why cells with more genes expressed (>3,500 genes) were also filtered out. Sometimes, cells with unexpectedly high counts and a large number of detected genes may represent doublets. But this is not always true depending on sequencing protocols,

batches, and cell types. Different cell types have huge variations for the number of genes detected. Also, what does ">5% of genes mapping to the mitochondria" mean is unclear. Does it mean (a) >5% of expressed genes are mitochondria genes or (b) >5% of mapped reads came from mitochondria?

(3) Page 5 (Line 28-30): A high fraction of mitochondrial counts are indicative of cells whose cytoplasmic mRNA has leaked out through a broken membrane. For cells with high % of mitochondria reads should be removed in QC step. Since gene expression levels calculated by RNA-seq are relative numbers, they should be normalized and scaled by total non-mitochondrial genes. It is unclear how gene expressions can be normalized by mitochondria genes.

(4) PCAs were used for t-SNE analysis. However, the choice of top PCs are fairly random and are not consistent: Page 5 Line 33: 13 PCs were selected; Page 6 Line 9: 14 PCs were selected; Page 6 Line 21: 7 PCs are selected. The top PCs selection should either have specific reason for each study or be consistent for all studies.

(5) Page 7 Line 8: p-values was determined by t-test. The basic assumption of using t-test is that "in-group" is very homogeneous and the variations follow normal distribution. Since the data is "log-transformed fold change", other statistical methods, such as Wilcoxon signed-rank test, will be more appropriate.

(6) Page 6 Line 18-19: "Cells were filtered based on $UMI < 10^6$, to exclude doubles/triplets". Low number of UMI are mainly due to PCR bias. This is because amplification of low amount of RNAs can result in substantial bias towards to certain fragments. Low UMI has nothing to do with doubles/triplets. The only way to detect doubles/triplets from scRNA-seq data is to use complicated computational approaches, such as Scrublet (Ref: Scrublet: computational identification of cell doublets in single-cell transcriptomic data." Cell systems 8.4 (2019): 281-291.). In fact, "doubles/triplets" will cause an increased number of UMI due to "more cells".

(7) Page 5 Line 40: what method is used for adjusting p-values? Benjamini hochberg or Bonferroni?

RESPONSE TO REVIEWERS

We thank all reviewers for their comments. Based on the comments, we have substantially revised the manuscript, including multiple new experiments and new figures (new panels in Fig. 3, new figs. 4, 5, revised fig. 7, and new Suppl. Figs. S9, S10, S12, S13, S14, new Table 1 and Movies 1-5). Please note that based on the advise of ZFIN nomenclature committee, we have renamed *etv2:Gal4* line into *etv2^{Gt(2A-Gal4)ci32}*, abbreviated as *etv2^{ci32Gt}*, to reflect the fact that it is a gene trap construct which has interrupted *etv2* locus. We had previously submitted the description of the generation and characterization of the *etv2^{ci32Gt}* line as a separate manuscript which has been recently published (Chestnut and Sumanas, Dev Dyn 2019; doi: 10.1002/dvdy.130).

The specific comments by reviewers were addressed as follows.

Reviewer #1:

Specific comments:

Why was the *Etv2:gal4*, UAS:GFP rather than the *Etv2:GFP* or *Etv2:mCherry* used for the scRNA-seq experiments? Why not do an *ETV2-2A-GFP* genomic insertion? Is there an advantage in using the *gal4/UAS* transgenic system? How accurate and stringent is *Etv2* expression represented by GFP in this model? In addition to the immunofluorescent data presented, was any molecular analysis performed? What is the half-life of GFP in this transgenic model? As judged by *etv2* expression in single cell RNA-seq data (fig 1 c), only a low fraction of the cells analyzed do express *etv2*.

Response: As we reported recently in a separate manuscript (Chestnut and Sumanas, Dev Dyn 2019; doi: 10.1002/dvdy.130) which describes generation and characterization of the *etv2^{ci32Gt}* gene-trap line, previously generated *etv2* reporter lines display some non-specific expression. We expected that the *gal4* knock-in line will recapitulate endogenous pattern of *etv2* expression more accurately, and will also allow to visualize *etv2*-deficient cells present in homozygous embryos. In addition, *Gal4/UAS* system allows for signal amplification and typically results in a brighter GFP fluorescence, compared with other approaches. We have performed detailed characterization and comparison of GFP fluorescence with endogenous *etv2* mRNA expression in a separate study (Chestnut and Sumanas, Dev Dyn 2019) and showed that GFP expression pattern recapitulates well the endogenous *etv2* expression. Reported half-life of EGFP is 26 hours, therefore GFP fluorescence is expected to stay much longer even after *etv2* mRNA expression has been downregulated. Our previous research has suggested (Sumanas et al 2008, Blood 111, 4500-4510; Glenn et al 2014, Dev Biol 393, 149-159) that *etv2* is expressed in early hematopoietic progenitors but its expression is downregulated as they differentiate. This explains low levels of *etv2* mRNA expression in many other cell populations identified by scRNA-seq. In addition, due to “dropouts” which happen when preparing libraries for RNA-seq, only a fraction of endogenous transcripts are present in the scRNA-seq libraries. As a result, cells that have low expression of *etv2*, may show 0 transcripts in scRNA-seq data which explains apparent absence of *etv2* in some cells.

Comment: In the scRNA-seq experiment, 2049 GFP+ cells were sequenced from 100-150 het embryos and 588 GFP+ cells from 75-100 from homo embryos. It is not clear how representative the scRNA-seq data are of overall GFP+ embryonic populations. What are the frequencies of GFP+ cells in het and homo embryos? This becomes very important when reporting the proportions of identified subsets as shown in figure 1e for example. The authors state that: “the LPM, cardiomyocyte and skeletal muscle (myocyte) populations were greatly increased in the homozygous embryos” but is this increase only relative within the context of the single cell populations analysed? It is not clear that these conclusions can be extrapolated to the full embryo.

Response: During FACS sorting of GFP-positive cells from heterozygous and homozygous embryos, 30,984 GFP+ cells out of total 1.639×10^6 cells and 9,067 GFP+ cells out of 458,341 total cells were isolated from $etv2^{ci32Gt}$ heterozygous and homozygous embryos, respectively, resulting in the frequencies of 1.89% and 1.98% GFP+ cells in heterozygous and homozygous embryos, respectively. Thus, the relative number of GFP+ cells was similar in heterozygous and homozygous embryos. We have now included this data in the main manuscript text.

Comment: Page 8 lines 40-41: “The presence of GFP-positive cardiomyocytes in $etv2:gal4$ heterozygous embryos could be explained by the loss of one functional $etv2$ allele, which resulted in some $etv2+$ cells differentiating as cardiomyocytes.” What are the evidences that $Etv2+$ progenitors do not give rise to cardiomyocytes? Has lineage tracing with $Etv2$ been performed?

Response: GFP expression has not been observed in cardiomyocytes using any other $etv2$ transgenic reporter lines in wild-type background (Palencia-Desai et al 2011, Development 138, 4721-4732). In addition, $etv2$ is not expressed in the myocardial field and endogenous $etv2$ expression does not overlap with early myocardial markers in wild-type embryos. Although, to our knowledge, no lineage tracing of $etv2$ cells has been performed in zebrafish embryos as yet, this makes it very unlikely that $etv2+$ cells differentiate into cardiomyocytes in wild-type embryos. In the manuscript we merely suggest a likely explanation of our results; this study was not designed to test differentiation of $etv2$ progenitors into cardiomyocytes.

Comment: Page 3, lines 38-39: “ $Etv2$ function is required to actively repress alternative cell fates in multipotent mesodermal progenitors.” What are the evidences that these progenitors are multipotent and that $Etv2$ is expressed in these progenitors? Based on the scRNA-seq data, few LPM cells express $etv2$. If those represent progenitors, one might expect to find $etv2$ still expressed.

Response: Appearance of only few LPM cells that express $etv2$ can be explained by low level of $etv2$ expression in these cells. The drop out rate in scRNA-seq experiments is estimated at 80% or even higher due to many technical reasons, meaning that only 20% or less of mRNA transcripts are captured in each cell. For genes that are expressed at a low level, some or multiple cells will show no transcripts. This is also evident in $etv2$ expression in EC1 and EC2 populations (Suppl. Fig. S2); there are multiple EC1 / EC2 cells with no $etv2$ expression. Even expression of $cdh5$, a classical marker for EC populations, is absent in a significant number of EC1 and EC2 cells (Suppl. Fig. S2). It is expected that $gal4$; UAS system results in signal amplification, therefore even low level of $etv2$ expression can result in substantial GFP expression which helps to capture these cells during FACS sorting. We have confirmed overlap of GFP and LPM gene co-expression in Fig. 7 and Suppl. Fig. S14.

We suggest that these progenitors are multipotent because they can differentiate into alternative cell fates (skeletal muscle) in the absence of $Etv2$ function. We admit that we have not demonstrated that a progeny of a single cell can acquire different fates, a classical definition of multipotency. Therefore we softened the statement and removed multipotency claim from that sentence.

Comment: Page 9 lines 5-6: “A population of RBC cells was expanded in the $etv2:gal4$ homozygous embryos”. What is the basis for claiming that RBC cells expanded? Are there more RBCs in $Etv2:gal4+/-$ embryos? This should be supported by additional evidences.

Response: Indeed, expansion of RBCs has not been previously reported. Based on scRNA-seq data, there is a higher percentage of RBC cells in $etv2^{ci32Gt}$ homozygous embryos than in heterozygous, while the overall fraction of GFP+ cells is similar. To confirm this result, we used three different approaches. 1) We have recently published global RNA-seq analysis of $etv2^{ci32Gt}$ embryos (Chestnut and Sumanas 2019, Dev Dyn). We reanalyzed this data to investigate expression of RBC-specific genes. Indeed, multiple RBC-specific genes, including six different hemoglobins were upregulated in $etv2^{ci32Gt}$ homozygous embryos at 24 hpf (Suppl. Table S3). Note that this was a completely different analysis using global RNA-seq of embryos at 24 hpf stage compared to scRNA-seq at the 20-somite stage presented in the current study. 2) We performed differential expression analysis of RBC marker genes identified in scRNA-seq analysis between homozygous and heterozygous embryos. Several RBC-specific

genes including three different globins were upregulated in $etv2^{ci32Gt}$ homozygous embryos (Suppl. Table S4). 3) We could not directly count RBC cells in $etv2^{ci32Gt}$ homozygous embryos because we did not have the line crossed into RBC-reporter line to do this analysis in a timely manner. Instead, we used a previously validated $etv2$ MO to knock down $etv2$ in $gata1:dsRed$ reporter embryos, which express $dsRed$ in RBCs. Based on the cell counts of FACS-sorted fluorescent cells, $etv2$ knockdown embryos show a significant and reproducible increase in the number of $gata1:dsRed$ -positive cells (Suppl. Fig. S9). All of these data point to increased erythropoiesis in $etv2$ -deficient embryos. The nature and the mechanism of this phenotype will require further investigation.

Comment: Differentiation of $etv2+$ progenitors into skeletal muscle: it is not clear how prevalent this is? What is the frequency of $etv2+$ derived skeletal muscle relative to the overall number of skeletal muscle cells at the embryonic stage analyzed (figure 3, 4)? Is this a marginal event or a widespread phenomenon? The scl morpholino data are rather weak and would need additional molecular-based evidences to support the involvement of Scl in this process.

Response: We have observed 3 ± 2.5 GFP+ myocytes per embryo in $etv2^{ci32Gt}$ heterozygous embryos and 18 ± 5.4 GFP+ myocytes per embryo in homozygous embryos. We now included these numbers in the manuscript text and Fig. 3e. Based on scRNA-seq data (Fig. 1e), GFP+ myocytes represent 1.2% of total GFP cells in heterozygous and 3.9% of total GFP cells in homozygous embryos. It is challenging to estimate the total number of skeletal muscle cells. Nevertheless, it is clear that the number of GFP+ myocytes is relatively small compared to the total number of skeletal muscle cells. Muscle cell counts in $etv2^{ci32Gt+/-}$ embryos injected with scl MO are based on 20 randomly selected embryos which were analyzed by confocal microscopy in two independent experiments. Increase in GFP+ myocytes was reproducible and highly statistically significant ($p < 0.01$). Control uninjected embryos from the same batch were analyzed in parallel. Scl MO-injected embryos were morphologically normal and showed previously reported defects in hematopoiesis and vascular development. To confirm that the analyzed cells are indeed skeletal muscle, we analyzed muscle actin reporter $actc1b:GFP$ expression in $etv2^{ci32Gt}$; UAS:mCherry embryos injected with scl MO. We show that these cells are positive for $actc1b:GFP$ expression (Fig. 3o-q), thus confirming that scl knockdown results in increased muscle differentiation of $etv2$ reporter cells.

Comment: Page 10, lines 31-32: "Muscle-specific myod expression was greatly inhibited upon $etv2$ overexpression". This does not strengthen the conclusion given that it is well established that $etv2$ overexpression in many cell types will push cells toward a vascular fate. Lack of myod expression only reflect a lack of skeletal muscle cells but does not directly implicate $etv2$ as a repressor of skeletal muscle fate.

Response: The data in Fig. 3t show that $etv2$ overexpression results in a partial loss of myod expression. We now confirmed this result by qPCR analysis (Fig. 3u). We agree that there could be different interpretations of this result possible. Indeed, $etv2$ overexpression will drive many cells towards vascular fate. However, myod-expressing cells are positioned in the somites and normally are exposed to the signals which induce muscle fate in wild-type embryos. It is not well understood why activation of endothelial program in these muscle progenitors prevents these cells from also initiating myogenesis in response to muscle inducing signals. Previous work by Org et al (EMBO J 34, 759-777 (2015) has demonstrated that Scl can occupy cardiac enhancers preventing their activation. We propose that $Etv2$ may function through or together with Scl to inhibit muscle development through a similar mechanism. However, inhibition of muscle expression of $etv2$ -overexpressing embryos could also be indirect. Further studies would be needed to identify the mechanism of how muscle differentiation is inhibited in $etv2$ -expressing cells.

Comment: In the $dkk1$ treated embryos (fig 4), what happens globally to myocytes? Are they reduced to the same extend as the GFP+ myocytes? The authors state that "Wnt signaling promotes muscle differentiation in multipotent $etv2+$ progenitors." Is this unique to the $etv2+$ derived myocytes?

Response: Previous study by Martin and Kimelman, 2012 (Dev Cell 22, 223-232) showed that heat-shock induction of *dkk1* expression at the 8-somite stage results in reduced myocyte differentiation which is limited to the tail region because Wnt signaling is required for myocardial differentiation of tailbud-derived multipotent progenitors during tail extension. We also observed similar reduction in myod expression in the tail region (Suppl. Fig. S13a,b). Because the heat-shock is performed at the 8-somite stage and some additional time is needed for induction of *Dkk1* expression, it is not expected that *Dkk1* would inhibit already established somites in the anterior and middle portions of the trunk. Wnt role in myocyte differentiation in multipotent progenitors in the tailbud has been previously demonstrated (Martin and Kimelman, 2012, Dev Cell 22, 223-32). Clearly, not all tailbud progenitors in the tailbud are *etv2*+ positive, and *etv2* is not expressed in muscle progenitors in wild-type embryos, therefore Wnt role would not be unique to the *etv2*+ derived myocytes.

Comment: Page 14, lines 43-44: “our work has demonstrated that cells in the vascular endothelial lineage arise from multipotent progenitors in the lateral plate mesoderm which can differentiate into skeletal muscle in the absence of *Etv2* function”. To fully demonstrate this, lineage tracing would be required.

Response: We revised this phrase to “suggested”

Minor comments

Page 10, lines 10-11: “Double heterozygous *etv2gal4/ci33* embryos displayed a dramatic increase in GFP+ myocytes compared to *etv2:gal4+/-* embryos”. It looks like a 3 to 4 fold increase in GFP+ myocytes which is far from “dramatic”. Same remark for the sentence: “SU5402-treated embryos showed a dramatic decrease in the number of ectopic GFP+ muscle cells”, data in which GFP+ myocytes decrease about two-fold.”

Response: These phrases were revised.

Comment: Page 10, lines 23-24: “The number of myocytes was significantly increased in the *etv2:gal4+/-* embryos injected with the previously validated *scl* morpholino (MO)”. Does this sentence refer to the overall number of myocytes per embryos?

Response: No, this refers to *etv2^{ci32Gt}* - expressing myocytes. We revised this sentence to avoid confusion.

Comment: Figure 5: it is very difficult to see co-expressing cells. Close-up images should be shown.

Response: Higher magnification images have been included (currently Fig. 7)

Comment: Page 14, lines 10-11: “previous studies have demonstrated the existence of a common vascular and endothelial progenitor, the hemangioblast”. Is that not rather a common hematopoietic and endothelial progenitor?

Response: We apologize, this error has been corrected.

--

Reviewer #2:

Major comments:

1. The authors need to find a way to isolate the *etv2* positive cells and they employ a transgenic line that has *gal4* inserted into the coding part of the gene. As a result, they use the heterozygous and homozygous cells of this line and treat the heterozygous as almost wild type. This can lead to wrong conclusions, especially if the authors don't show in detail that heterozygosity of *etv2* does not disturb main functions. Another way to solve the problem would be to use the Tg(-2.3 *etv2*:GFP) line, perform

scRNA-seq and compare it to their current results. It is understandable that these two lines do not mark exactly the same cell populations, but some commonalities should be present. In addition, the authors claim that the phenotype of their transgenic animals is more severe than the one of the actual mutants. This may be caused by deletion of regulatory elements that are located close to *etv2* gene. Can the authors check the expression of these adjacent genes? Such a problem leads to an even more urgent need to sequence cells from the Tg(-2.3 *etv2*:GFP) or any other transgenic that could verify the current results.

Response: We have performed detailed characterization of vascular development in the *etv2*^{ci32Gt} line and previously generated TgBAC(*etv2*:GFP) and Tg(-2.3 *etv2*:GFP) lines and we did not find significant differences in vascular development between *etv2*^{ci32Gt} heterozygous embryos and the other reporter lines (Chestnut and Sumanas, Dev Dyn 2019; doi: 10.1002/dvdy.130). The two differences that we have observed include the presence of a few GFP+ skeletal muscle cells and also a cluster of apoptotic cells observed in *etv2*^{ci32Gt+/-} embryos which we attribute to the partial loss of *etv2* function, and this is discussed in the current manuscript. We did perform single-cell RNA-seq of Tg(-2.3 *etv2*:GFP) embryos at similar embryonic stages using Fluidigm approach (Fig. 8). Despite the difference in sequencing platforms, we have identified in Tg (-2.3 *etv2*:GFP) embryos many of the same cell populations with the same marker genes, including endothelial progenitor cells, vascular endothelial cells (three different subpopulations were observed in Tg(-2.3 *etv2*:GFP) embryos likely due to better separation of marker genes because of greater sequencing depth), red blood cells, macrophages and presumptive tailbud progenitors. Cell populations which were observed in *etv2*^{ci32Gt} line but not identified in Tg(-2.3 *etv2*:GFP) embryos include cardiomyocytes, skeletal muscle cells, apoptotic cells and LPM progenitors. We did not observe any GFP expression in cardiomyocytes, skeletal muscle cells or apoptotic-looking cells in Tg(-2.3 *etv2*:GFP) or TgBAC (*etv2*:GFP) embryos, therefore the presence of these populations is a likely consequence of partial loss of *etv2* function in *etv2*^{ci32Gt} embryos. *Etv2* inhibition is known to result in EC apoptosis (Pham et al., 2007, Dev Biol 303, 772-783), and we have previously demonstrated that *etv2*-positive cells can differentiate into cardiomyocytes in *etv2*-inhibited embryos (Palencia-Desai et al., 2011, Development 138, 4721-4732). Regarding LPM progenitors, this population was not identified in Tg(-2.3 *etv2*:GFP) embryos either because fewer cells were sequenced in this line and low number of LPM cells could not be separated into a distinct cluster, or because this population may have low GFP expression as these cells may have just initiated *etv2* expression. *etv2*^{ci32Gt} line is significantly brighter than Tg(-2.3 *etv2*:GFP) line, likely due to the amplification of GFP expression by Gal4:UAS system, therefore low GFP cells may be missed in *etv2*:GFP line. We have now included the comparison of the sequencing results between the two different platforms in the Discussion section.

Comment: 2. The number of heterozygous and homozygous cells sequenced differs greatly between them. This maybe a problem in comparing these two experiments and identifying missing or enriched populations. Can the authors equilibrate these differences or prove that they do not affect the results?

Response: During bioinformatical analysis GFP+ cells from heterozygous and homozygous *etv2*^{ci32Gt} embryos were combined for clustering. After clustering, cells were then plotted on separate graphs based on their genotype. Thus they were analyzed together in the same experiment.

Comment: 3. The authors claim that Wnt and FGF signaling promote skeletal muscle fate. Can they show in their scRNA seq data that genes regulated by these pathways have altered expression?

Response: Wnt and FGF pathways can induce different effector genes depending on the cell type and signaling context. Myogenic regulator factors MyoD and Myf5 are known to be transcriptionally activated by Wnt signaling (Tajbakhsh S, et al. Differential activation of Myf5 and MyoD by different Wnts in explants of mouse paraxial mesoderm and the later activation of myogenesis in the absence of Myf5. Development. 1998;125:4155–4162; Chen AE, et al. Protein kinase A signalling via CREB controls myogenesis induced by Wnt proteins. Nature. 2005;433:317–322). In our scRNA-seq results, zebrafish homologs *myod1* and *myf5* are among the top markers expressed in zebrafish myocytes, and the number of myocytes is greatly increased in homozygous *etv2*^{ci32Gt} embryos.

It has been similarly demonstrated that FGF signaling induces *myf5*, *myod* and *msgn1* expression in zebrafish embryos (Row et al. BMP and FGF signaling interact to pattern mesoderm by controlling basic helix-loop-helix transcription factor activity. *eLife* 2018; 7: e31018). Thus although we do not see global upregulation of Wnt and FGF targets, we see upregulation of specific myogenic factors, known to be controlled by these signaling pathways.

Comment: 4. Can *scl* overexpression rescue the phenotype?

Response: We performed a rescue experiment by injecting *scl* mRNA into heterozygous and homozygous *etv2*^{ci32Gt} embryos. In either case, no significant change in the number of GFP+ myocytes was observed (data not shown). This suggests that *scl* mRNA may not function downstream of *etv2* in repressing muscle differentiation. We discuss these interpretations in the revised manuscript. A caveat is that it is not possible to exclude other technical reasons why *scl* mRNA failed to rescue the phenotype. Although we confirmed *scl* mRNA activity by the expansion of hematovascular marker expression, it is possible that injected mRNA does not stay long enough to repress muscle differentiation.

Minor comments

Comment: 1. Page 7 line 30: Supp. Fig. S2 should be mentioned here.

Response: We are a bit puzzled about this comment because the description of the phenotype in *etv2*^{ci32Gt} embryos is shown in Supp. Fig S1 (not S2) while Suppl. Fig. S2 is discussed later in the text. Suppl. Fig. S2 shows violin plots for the same genes as in Fig. 1c (which only has t-SNE plots); we were unable to fit them all into a single figure.

Comment: 2. On Figure 1 the authors claim that they show also markers of erythroid cells and other populations (specifically in Fig 1c,d) but that is not so. Can the authors add all the markers that are mentioned in the text?

Response: We apologize, RBC markers are shown in Suppl. Fig. S5; we removed reference to Fig. 1c. Fig. 1d does show all cell types including RBCs. We had to compromise on how many different markers and different cell types we can show in the main Figure 1 and keep the panels still legible. Therefore we only chose to show a single marker for 8 different cell types in the Fig. 1c, and additional markers and cell types are shown in Suppl. Fig. S2-S8. Even then we limited two markers per cell type in supplemental figures. Adding additional markers could double the number of supplemental figures, and we feel that it is unnecessary. Instead, we added Table 1 to the manuscript which lists all top marker genes for all populations. This should make it easier to match the text description with the original results. In addition, Suppl. Table S1 shows expression of all marker genes in every cell type, while Suppl. Table S2 lists expression of any gene in all cell types. From this data, expression of any gene over all cell types can be easily queried.

Comment: 3. The authors claim that the homozygous transgenic animals have more erythroid cells. Can the authors verify this using an independent method like WISH or by FACS analysis of an erythroid reporter animal?

Response: Please see the response to reviewer 1. We analyzed results from RNA-seq performed using whole-embryo RNA from *etv2*^{ci32Gt} heterozygous and homozygous embryos (Chestnut and Sumanas, *Dev Dyn* 2019). Multiple erythroid-specific genes were increased in homozygous embryos. We further performed FACS analysis on *etv2* injected *gata1:dsRed* reporter embryos. A reproducible and statistically significant increase in erythroid cells was observed (Suppl. Fig. S9).

Comment: 4. The authors show results on heterozygous and homozygous animals, but never on real wild type animals. Can the authors include wild type animals in any case that is pertinent, for example in WISH experiments?

Response: Please see the response above to the question #1. The problem for the most experiments is that only knock-in embryos show GFP expression, while wild-type embryos are not positive for GFP and cannot be analyzed in the same way. As suggested, we performed a control whole mount in situ

hybridization using TgBAC(etv2:GFP) line, and a similar overlap with prrx1a and twist1a expression was observed (Suppl. Fig. S14).

Comment: 5. Page 8 Line 16,17. Apoptosis and cell cycle genes do not exist in Fig 1c,d.

Response: We apologize, the reference to Fig. 1c was removed. We added Table 1 which lists top marker genes for each population; Fig. 1d does include several marker genes for each cell population including apoptotic population.

Comment: 6. Page 10 line 2 myod expression is missing from the figure.

Response: This section has been revised. A new Fig. 5 which shows overlap between GFP and myod expression has been added.

Comment: 7. Can the authors clarify how they count the GFP+ myocytes?

Response: This was added to the methods section.

--

Reviewer #3:

The central premise of this work is that etv2-expressing cells in the LPM make a fate decision controlled by etv2/ scl function relative to Wnt and Fgf activity. An important consideration when making this assumption is whether there is an effect on progenitor cells in the LPM that may affect how they respond to signals from their environment. It was not possible to discern from the time-lapse movies how the GFP+ progenitor cells migrate in etv2 heterozygotes and whether this is altered. If they show a different spatial organisation prior to or during migration they are likely to experience different cues that affect their specification. In order to address this point the authors should carefully characterise etv2-expressing cells in heterozygotes and homozygous animals using dorsal and lateral views to highlight cell movement to answer whether cells migrate differently in homozygous animals. A similar point holds for animals in which Wnt and Fgf are altered.

Response: We performed new time-lapse imaging using lateral and dorso-lateral orientations (Fig. 4 and Suppl. Fig. S12, Movies 1-5). We did not see differences in GFP+ cell localization prior to the 8-somite stage (Compare Fig. 4a and g, also Suppl. Fig. S12a and g, and data not shown). Starting at about 8-10-somite stages, GFP+ cells in etv2^{ci32Gt} heterozygous embryos migrate from the LPM in-between the somites to the midline where they coalesce into axial vasculature, similar to the migration previously observed with other GFP reporter lines (Kohli et al 2013, Dev Cell 25, 196-206). However, some GFP+ cells in etv2^{ci32Gt} heterozygous embryos and many cells in homozygous embryos can be observed migrating into the somites where they elongate and differentiate into the skeletal muscle (Fig. 4, please see higher magnification image of such transdifferentiating cell in Fig. 4m and Movie 4). Heat-shock for the induction of Dkk1 and dnFGFR1 induction was performed at the 8-somite stage. Therefore this could not affect the initial positions of GFP+ cells in the LPM at or prior to 8-somite stage.

Comment: I was also not clear if there are changes to the relative number of GFP+ cells at different stages of development in both genotypes or in when Fgf or Wnt are inhibited as this may also explain why there are more or fewer GFP+ myofibres. Careful quantification of cell number relative to their position is needed to address this issue.

Response: As described above, similar percentage of GFP+ cells was observed in etv2^{ci32Gt} heterozygous and homozygous embryos. To test if HS:Dkk1 or HS:dnFGFR1 affected overall number of etv2-reporter cells, we performed FACS sorting of fluorescent reporter cells. No significant change in the number of etv2-reporter cells was observed in different transgenes (Suppl. Fig. S13c).

Comment: Given that the entire paper relies on the new etv2 line I was surprised that this is not described in full in this paper. It is important to do this as I had the impression that the expression differs

from a previously published line made with a BAC and it is not clear in this work how accurately the new CRISPR-generated line expresses GFP relative to *etv2*.

Response: We had decided that it would be beneficial to publish characterization of this line as a separate manuscript. We had originally included a set of figures for reviewers only (as a separate supplemental file) which showed characterization of *etv2*^{ci32Gt} line; we apologize if this was missed and was not made clear. By now the *etv2*^{ci32Gt} characterization paper has been published and should provide all the details to answer this question (Chestnut and Sumanas, Dev Dyn 2019; doi: 10.1002/dvdy.130).

Comment: A similar caveat holds for descriptions for cell behaviour - it is simply not possible to see the different cell populations described in the text from the figures shown, specifically how the LPM progenitors migrate. Clearer figures demonstrating the spatial distribution of GFP+ cells is needed to clarify how this may affect their fate switch in an absence of *etv2* function.

Response: We reimaged the embryos and provide higher magnification images with additional labeling which hopefully answers this point (Fig. 4 and Suppl. Fig. S12, Movies 1-5).

Comment: The hypothesis that *etv2* acts to inhibit myogenesis is tested by injecting RNA for *etv2* and assaying *myod* expression. A striking result is shown indicating a unilateral loss of *myod* expression at early somite stages. This should be corroborated by qPCR for *myod* and other myogenic genes including *myf5*.

Response: We performed qPCR for *myf5*, *myod* and *myog* as suggested which confirmed reduction of their expression in *etv2*-overexpressing embryos (Fig. 3u).

Comment: The statement that GFP+ cells are forming myofibres should be tested by antibody staining to show that these express markers of differentiated myofibres.

Response: Expression of GFP in slow and fast myofibers has been analyzed using specific antibodies. This confirmed an overlap of GFP with antibody staining in fast (but not slow) myofibers (Suppl. Fig. S10).

Comment: One important piece of evidence for the role of *etv2* in dictating fate of this early LPM progenitor population is to show where progenitor cells switch to a myogenic fate and when this occurs. This could be accomplished by examining *myod* expression relative to GFP+ cells in *etv2* mutants at different stages of development to show if there is ectopic expression in GFP+ cells and when this occurs.

Response: As suggested, we examined GFP+ expression in *etv2*^{ci32Gt} embryos at earlier stages of 5 and 8-somites. At the 5-somite stages, majority of embryos did not show overlapping *myod* and GFP co-expression (data not shown). However, at the 8-somite stage GFP and *myod* co-expressing cells were apparent at the ventral portion of the somites (Fig. 5).

Reviewer #4:

Major comments:

(1) Over-expression of *Etv2* have been extensively studied. It will be interesting to compare this study (*Etv2*-deficient) with published *Etv2*-overexpression datasets, and see the overlap (e.g., whether key upregulation markers in this study are downregulated in *Etv2*-overexpression datasets).

Response: In this study, we compared *etv2*^{ci32Gt} heterozygous and homozygous embryos at a single cell level, while previous studies have used global transcriptome analysis. Key marker genes for several cell populations identified in our study, including muscle cells and cardiomyocytes have not been previously reported to be downregulated in *Etv2*-overexpression datasets. However, analysis has not been performed at the same stage. Our previous microarray study on *etv2* overexpressing embryos was

performed at the tailbud stage, when muscle or cardiomyocyte genes are not expressed yet (Wong et al., Dev Dyn 2009, 238, 1836-50).

(2) Page 5 (Line 17-19): To perform QC, it is reasonable to remove cells with low number of genes expressed (<200). But it is unclear why cells with more genes expressed (>3,500 genes) were also filtered out. Sometimes, cells with unexpectedly high counts and a large number of detected genes may represent doublets. But this is not always true depending on sequencing protocols, batches, and cell types. Different cell types have huge variations for the number of genes detected. Also, what does “>5% of genes mapping to the mitochondria” mean is unclear. Does it mean (a) >5% of expressed genes are mitochondria genes or (b) >5% of mapped reads came from mitochondria?

Response: During the quality control of 10X single-cell RNAseq data (merged from the heterozygous and homozygous embryos), we analyzed the distribution of the number of genes expressed in individual cells. The median nGene count for cells was 1599.5 and the mean was 1555.09. We removed 9 cells expressing over 3,500 genes, which appeared to be outliers in the dataset and are likely to be doublets/multiplets. The removal of doublets/multiplets which may exhibit an abnormally high gene count is recommended by Satija et al. in their standard pre-processing workflow to eliminate potential doublets/multiplets. (see ‘Standard pre-processing workflow’ on https://satijalab.org/seurat/v3.1/pbmc3k_tutorial.html). However, we do agree that there is a possibility that these cells are not doublets but are in fact single cells that express a very high number of genes; this would have to be determined using a more advanced algorithm to detect doublets (eg. Doublet decon, Scrublet) as suggested by the reviewer. In the event that those 9 cells are determined to be single cells, it is unlikely that the inclusion of these cells to the dataset would have an impact on downstream clustering and the overall results of this study.

As part of the quality control steps, we removed cells that had more than 5% of counts that mapped to the mitochondrial genome. We have rephrased the sentence to “.. **or >5% of counts mapping to the mitochondrial genome**”.

(3) Page 5 (Line 28-30): A high fraction of mitochondrial counts are indicative of cells whose cytoplasmic mRNA has leaked out through a broken membrane. For cells with high % of mitochondria reads should be removed in QC step. Since gene expression levels calculated by RNA-seq are relative numbers, they should be normalized and scaled by total non- mitochondrial genes. It is unclear how gene expressions can be normalized by mitochondria genes.

Response: During the scaling of Normalized data, we ‘regressed out’ cell-cell variation in gene expression driven by mitochondrial gene expression and the number of detected molecules. (see ‘Scaling the data and removing unwanted sources of variation’ https://satijalab.org/seurat/v2.4/pbmc3k_tutorial.html). We have corrected the error and re-phrased this sentence to “**Prior to dimensionality reduction, a linear transformation was performed on the normalized data. Unwanted cell-cell variation driven by mitochondrial gene expression and the number of detected molecules (nUMIs) was removed by regressing out these variables during the scaling of data.**”

(4) PCAs were used for t-SNE analysis. However, the choose of top PCs are fairly random and are not consistent: Page 5 Line 33: 13 PCs were selected; Page 6 Line 9: 14 PCs were selected; Page 6 Line 21: 7 PCs are selected. The top PCs selection should either have specific reason for each study or be consistent for all studies.

Response: In order to select the top principal components (PCs) that explain the majority of the variance in the data, we adopted 3 different methods – PCHeatmap, JackstrawPlot and ElbowPlot, as outlined by Satija et al. (see ‘Perform linear dimensional reduction’ on https://satijalab.org/seurat/v2.4/pbmc3k_tutorial.html). For the merged data from the homozygous and heterozygous embryos, the top 13 PCs were selected based on the information obtained from the 3 methods above. In the analysis of the etv2 heterozygous dataset alone, the top 15 PCs appeared to explain the majority of variance in the data and for the subset of Endothelial cells (page 6, line 6/7), PCA

was performed again; here, the top 14 PCs were selected based on the above methods. For pseudotime analysis in Monocle, a subset of populations from the merged data was used – LPM, EPCs, EC1, EC2 and Myocytes. Once again, the top PCs were determined using the `plot_pc_variance_explained()` function in Monocle, which outputs a graph similar to an Elbowplot. Based on the graph, the top 7 PCs were selected for downstream clustering. We performed multiple iterations of clustering on the datasets, using different numbers of PCs and based on the results picked the optimal number of PCs. It is important to mention that using a higher number of PCs than the numbers selected above did not change the overall clustering of the data. We understand, however, that it is possible to select an arbitrary number of PCs (eg. 15 or 20) and perform clustering analysis on all datasets using a constant number of PCs. A potential, albeit rare, complication with this approach would be if the number of significant PCs of a dataset exceeds this constant value; in which case, we would unintentionally omit certain PCs that would have a significant impact on the clustering results.

(5) Page 7 Line 8: p-values was determined by t-test. The basic assumption of using t-test is that “in-group” is very homogeneous and the variations follow normal distribution. Since the data is “log-transformed fold change”, other statistical methods, such as Wilcoxon signed-rank test, will be more appropriate.

Response: We agree with the reviewer’s comments and have performed two-tailed Mann-Whitney U test which should be equivalent to a Wilcoxon rank-sum test. The suggested Wilcoxon signed-rank test couldn’t be used because it works only for data sets with equal numbers of samples. Revised p values are shown in Suppl. Fig. S6.

(6) Page 6 Line 18-19: “Cells were filtered based on $UMI < 10^6$, to exclude doubles/triplets”. Low number of UMI are mainly due to PCR bias. This is because amplification of low amount of RNAs can result in substantial bias towards to certain fragments. Low UMI has nothing to do with doubles/triplets. The only way to detect doubles/triplets from scRNA-seq data is to use complicated computational approaches, such as Scrublet (Ref: Scrublet: computational identification of cell doublets in single-cell transcriptomic data." Cell systems 8.4 (2019): 281-291.). In fact, “doubles/triplets” will cause an increased number of UMI due to “more cells”.

Response: In the pseudotime analysis workflow used in Monocle, low-quality cells are filtered based on recommended steps outlined by Trapnell et al. (see ‘Filtering low-quality cells’ on <http://cole-trapnell-lab.github.io/monocle-release/docs/#filtering-low-quality-cells-recommended>). As the reviewer points out, it is only cells with a UMI count above $1e6$ that should be excluded; i.e only cells with a UMI count below $1e6$ are included for further analysis. We have corrected the error in line 18-19 Page 6 and describe further filtering performed on UMI counts. **“ Cells with a UMI count > $1e6$ were excluded from the data to exclude potential multiplts. Furthermore, upper and lower bounds on nUMI were set at 2 standard deviations above and below the mean UMI to remove low-quality cells. Only cells falling within these boundaries were included for further downstream analysis.”** It is important to note that when performing pseudotime analysis in Monocle, we subset populations from a Seurat object to create a new Seurat object and then convert it to a CellDataSet object (which is the data structure implemented in Monocle). Thus, although we subject the data to filtering in Monocle, these cells have already been filtered based on the Seurat quality control and preprocessing workflow.

(7) Page 5 Line 40: what method is used for adjusting p-values? Benjamini hochberg or Bonferroni?

Response: The differential expression tests in Seurat apply Bonferroni correction to return adjusted p-values.

Reviewers' Comments:

Reviewer #1:

Remarks to the Author:

The authors have addressed all my concerns.

Reviewer #2:

Remarks to the Author:

The authors have substantially addressed my comments and the manuscript is greatly improved.

Reviewer #4:

Remarks to the Author:

The authors have addressed all of my questions.